**Subject Category:**
Biology (whole organism)

physiology/ecology/behaviour

horsefly, host choice, blood-seeking, thermoreception, polarization vision, parasite–host interaction

**Author for correspondence:**
Gábor Horváth
e-mail: gh@arago.elte.hu

# Attractiveness of thermally different, uniformly black targets to horseflies: *Tabanus tergestinus* prefers sunlit warm shiny dark targets

Gábor Horváth[1], Ádám Pereszlényi[1,3], Tímea Tóth[1], Szabolcs Polgár[1] and Imre M. Jánosi[2,4]

[1]Environmental Optics Laboratory, Department of Biological Physics, and [2]Department of Physics of Complex Systems, ELTE Eötvös Loránd University, 1117 Budapest, Pázmány sétány 1, Hungary
[3]Hungarian Natural History Museum, Department of Zoology, Bird Collection, 1083 Budapest, Ludovika tér 2-6, Hungary
[4]Max Planck Institute for the Physics of Complex Systems, Nöthnitzer Strasse 38, 01187 Dresden, Germany

GH, 0000-0002-9008-2411

From a large distance tabanid flies may find their host animal by means of its shape, size, motion, odour, radiance and degree of polarization of host-reflected light. After alighting on the host, tabanids may use their mechano-, thermo-, hygro- and chemoreceptors to sense the substrate characteristics. Female tabanids prefer to attack sunlit against shady dark host animals, or dark against bright hosts for a blood meal, the exact reasons for which are unknown. Since sunlit darker surfaces are warmer than shady ones or sunlit/shady brighter surfaces, the differences in surface temperatures of dark and bright as well as sunlit and shady hosts may partly explain their different attractiveness to tabanids. We tested this observed warmth preference in field experiments, where we compared the attractiveness to tabanids (*Tabanus tergestinus*) of a warm and a cold shiny black barrel imitating dark hosts with the same optical characteristics. Using imaging polarimetry, thermography and Schlieren imaging, we measured the optical and thermal characteristics of both barrels and their small-scale models. We recorded the number of landings on these targets and measured the time periods spent on them. Our study revealed that *T. tergestinus* tabanid flies prefer sunlit warm shiny black targets against sunlit or shady cold ones with the same optical characteristics. These results support our new hypothesis that a blood-seeking female tabanid prefers elevated temperatures, partly because

her wing muscles are more rapid and her nervous system functions better (due to faster conduction velocities and synaptic transmission of signals) in a warmer microclimate, and thus, she can avoid the parasite-repelling reactions of host animals by a prompt take-off.

# 1. Introduction

Cattle annoyed by tabanids grow thinner, thus their meat and milk production decreases [1]. Tabanid attack and annoyance result in health problems of livestock and economical losses for horse- and cattle-keepers [2,3]. Thus, there is a great need for effective tabanid traps for tabanid control. In order to design such traps, it is important to know the reasons for host attractiveness to tabanids.

The attractiveness of host animals to tabanids has several different reasons, such as feeding on vertebrate secretions (e.g. sweat, tears and blood as nutrition sources), or the use of the host's body surface as a basking or swarming site [4]. From a large distance, tabanids find their host animals by means of different cues, such as the radiance (brightness/darkness and/or colour and/or contrast) [5–11] and the degree of polarization [12–18] of host-reflected light, shape and size [5,8,9,19,20] and odour [21–27]. The role of the host's movement in host finding by tabanids is equivocal, because depending on the target's velocity, some species prefer moving targets, others are not influenced by the target's motion [5,9,28,29]. After alighting on the body surface of a host, female tabanids decide whether they remain on it to (i) look for vessels for blood-sucking or (ii) rest (bask, for instance). Being on the host's body surface, tabanids use their mechano-, thermo-, hygro- and chemoreceptors to test the physical and physiological characteristics of the substrate [4,30]. These receptors are most effective if the fly is in contact with the host's body surface. Therefore, mainly odour and vision are used to locate hosts from a distance. However, tabanids may decide to land or not to land already in close vicinity of the target without contact. The thermoreceptors in tabanids (as usually in Diptera) are in the legs, antennae and mouthparts [4,31,32].

Female tabanids prefer to attack sunlit against shady dark (brown, black) host animals, or dark against bright (grey, white) hosts for a blood meal [16,33–35]. A qualitative observation by Tashiro & Schwardt [36] showed that tabanids attack dark cattle more frequently than white. In a quantitative field experiment, the attractiveness of sunlit brown horses to tabanids was about four times larger than that of sunlit white ones [33]. That is why the most effective tabanid traps use black decoy targets [8,19,37–41].

Field experiments and observations have shown that tabanids prefer sunlit dark hosts/targets [16,34,42]. Horse- and cattle-keepers know well that tabanids practically do not follow their host animals to the shade of stables or forests, thus hosts can escape from tabanid attacks into shady regions. Horváth et al. [33] found that in comparison with a white horse, a brown horse spent 2.2 times longer period in a tabanid-free shady forest than in the sunny field with intense tabanid attacks. Using video recordings of tabanids in the field, Caro et al. [43] could study the behaviour of horseflies only around sunlit zebras and horses, because tabanids practically do not attack shady host animals. Otártics et al. [44] found that H-traps in sunny places caught significantly more female tabanids than at shady sites.

The tabanid preference of sunlit dark hosts/targets is rather enigmatic, because in the case of a given host species, the blood quality (with respect to the egg production in female tabanids) of darker hosts should practically be the same as that of brighter ones. Similarly, the mechanical (skin, hair) and chemical (carbon dioxide, odour) characteristics of the body surface of darker and brighter hosts of the same species cannot be very different. A relevant difference can be, however, that sunlit animals could sweat more than shady ones (especially if a sunlit dark animal is compared with a shady bright one). A more sweating host animal may have a bacteria population growing differently compared to a less sweaty host, which can thus influence chemicals released by the host's skin.

According to a widespread explanation, the responsiveness of blood-sucking insects (including tabanids) to targets showing strong radiance contrast with the background is not surprising as large homeothermal host animals as blood sources are low-radiance objects that will appear dark in contrast with high-radiance vegetation [45]. Therefore, host-seeking insects (together with tabanids) prefer to land on dark, low-radiance colours similar to those of many host animals [4]. However, the explanation of the attraction of tabanids by dark hosts is more complex, because bright brown and grey ungulates (e.g. horses, cattle, antelopes) belong also to the hosts of tabanids. Apart from darkness, the degree of polarization of target-reflected light plays also an important role: a dark target

is more attractive if it is shiny, thus strongly polarizing, because matt black surfaces, with only a weak polarizing ability, are much less attractive to tabanids than shiny ones [15,33]. Furthermore, high degrees of polarization of reflected light help tabanids to select sunlit dark host animals from the dark patches of their visual environment [13].

In this work, we will test the hypothesis that the thermal characteristics of (sunlit/shady) dark hosts play an important role in the host choice by tabanids. In sunshine, the surface features of dark and bright hosts of the same species differ mainly in temperature: sunlit darker surfaces are warmer than brighter ones, because the former absorb more sunlight than the latter. Therefore, our working hypothesis is that the differences in the surface temperature between dark and bright hosts can explain the differences in their attractiveness to tabanids landed (because from a distance, a tabanid cannot sense the host's surface temperature).

Tabanids prefer to congregate in enclosures that accumulate heat during daylight, notably motor vehicles [46]. Most tabanid species feed during full daylight and are often most in evidence on the hottest, sunniest days. Most species are exophilic and exophagic, an exception being the loiasis vector, *Chrysops silaceus*, which will enter houses to feed [4]. However, Bracken & Thorsteinson [28] found that black or white decoys simulating warmth of mammalian hosts were not significantly more attractive to tabanids than similar objects at ambient temperature. On the other hand, when a female tabanid has landed on a host, heat is also required to stimulate her blood-sucking [47]. Thus, the role of the surface temperature of the hosts in the host choice of tabanids is equivocal.

In previous studies, the attractiveness of thermally different but optically uniform targets to tabanids has not been compared. To fill this gap, we performed field experiments, in which the attractiveness to tabanids of a warm and a cold shiny black barrel (imitating dark hosts) with the same optical characteristics was compared. Using imaging polarimetry, thermography and Schlieren imaging, we measured the physical (optical and thermal) characteristics of both targets and their small-scale models. We recorded the number of landings on these targets and measured the time periods spent on them. Our study revealed the tabanid attractiveness of optically uniform dark host-mimicking targets versus the target's surface temperature. We showed that tabanids prefer to land on sunlit surfaces of elevated temperatures and they stay there much longer than on colder targets of equivalent optical properties. We formulate a hypothesis that such preference might be related to a thermal strategy where an optimal temperature of wing muscles can be a selective advantage.

# 2. Material and methods

## 2.1. Choice experiments

Our choice experiments 1–5 were performed on a horse farm in Göd (47°43′ N, 19°09′ E) in July 2018 from early forenoon to late afternoon of warm, sunny and windless days (experiment 1: 10 July, 09.50–17.30 = UTC + 2 h; experiment 2: 14 July, 10.30–18.00; experiment 3: 15 July, 10.30–18.00; experiment 4: 21 July, 10.00–17.30; experiment 5: 27 July, 10.00–14.00) where tabanid flies (*Tabanus tergestinus* Egger 1859) are abundant in summer, as we have experienced in our earlier field experiments [14,16,33,48,49]. Two shiny black plastic cylindrical barrels (height = 42 cm, diameter = 30 cm, wall thickness = 5 mm) with black caps were placed on white tetrapodal plastic stools (height = 46 cm) 1 m apart at a site where there was not any shade cast by an object (vegetation, building) during the experiments. We used barrels with a shiny black outer surface, because due to the rule of Umow [50], smooth black surfaces reflect light with the highest degree of polarization, and blood-seeking female tabanids are the strongest attracted to such polarized light.

The warm barrel contained only air which became and remained warm in sunshine, because its closed cap hindered the emanation of warm/hot air. The cold barrel was filled with tap water, into which 10 iceakkus (Aspico G40, 0.25 l, 0.76 kg plastic container filled with a gel of low temperature of congelation) frozen in a common kitchen-refrigerator were submerged. Its closed cap hindered the outflow of the cold water or the emanation of water vapour. Since in sunshine, the outer surface of the air-filled barrel became warm, while that of the water-iceakku-filled barrel remained relatively cold, the former and latter barrel are henceforward called 'warm' and 'cold', respectively. The positions of the warm and cold barrels were hourly inverted during each experiment. Since in experiment 1 the iceakkus were not refreshed, the outer surface temperature of the cold barrel gradually increased in time, and near the end of this experiment it approximated that of the warm barrel. To eliminate this

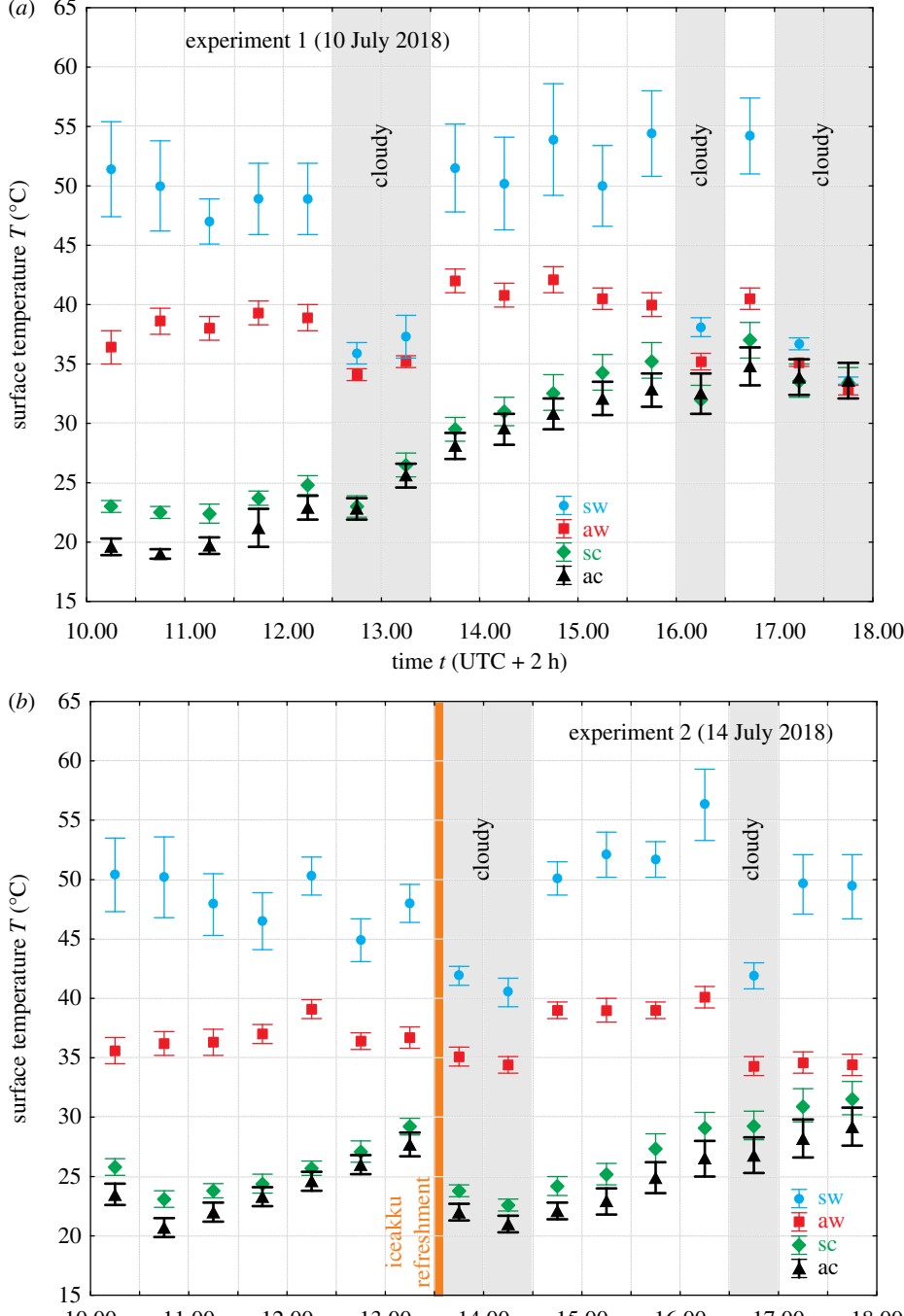

**Figure 1.** (a) Average (different symbols) ± s.d. (vertical bars) of the surface temperature T of the solar and antisolar sides of the warm and cold barrels in experiment 1 (on 10 July 2018) measured by thermography as a function of time t (local solar time = UTC + 2 h). At a given point of time, T was averaged in the rectangles on the barrel's shells shown in figure 2c,d. sw, solar side of the warm barrel; aw, antisolar side of the warm barrel; sc, solar side of the cold barrel; ac, antisolar side of the cold barrel. (b) As (a) for experiment 2 (on 14 July 2018) in which the iceakkus were refreshed in the cold barrel at 13.30 represented by a vertical orange line. Grey vertical bands show the intervals when the sun was occluded by clouds.

problem, in experiments 2–5, the iceakkus were refreshed in the cold barrel at half-time. Thus, we could ensure relatively great temperature differences between the warm and cold barrels (figure 1).

Since we did not want to lose the results of experiment 1, we used these data, although the temporal temperature variation of the water-filled cold barrel was different in the first experiment (without refreshing the iceakkus) compared to the other four experiments (with refreshing the iceakkus). The only irrelevant consequence of this was that in the second half of experiment 1, higher surface

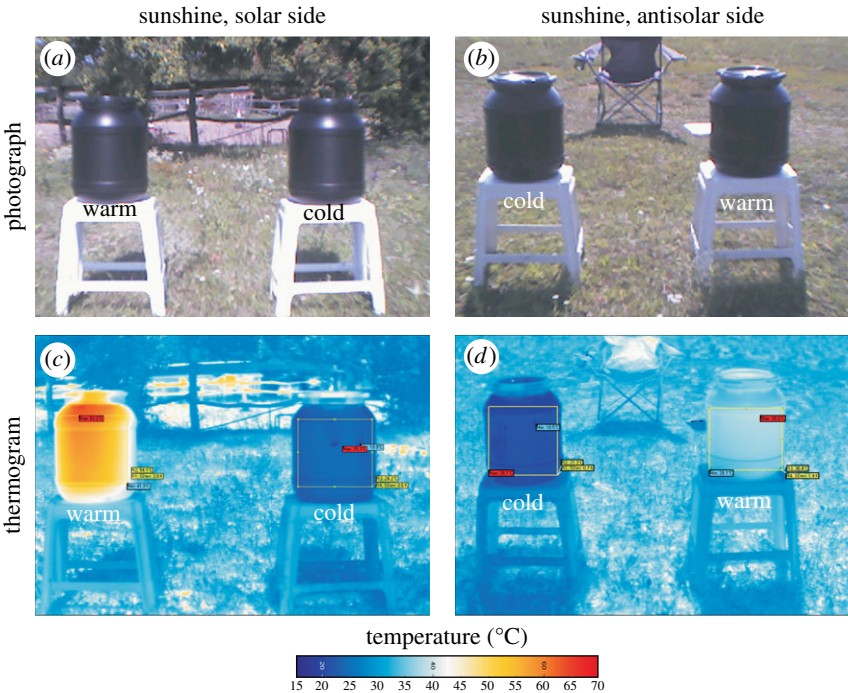

**Figure 2.** Photographs (*a,b*) and thermograms (*c,d*) of the solar (*a,c*) and antisolar (*b,d*) sides of the warm and cold black barrels used in the choice experiments measured by a thermocamera in sunshine at 09.50 (=UTC + 2 h) on 10 July 2018. The rectangles on the barrel's shells show the areas where the surface temperature $T$ of the barrels was averaged.

temperatures of the water-filled barrel occurred than in experiments 2–5. This, however, did not influence our results and conclusions: our main aim was to determine the number of landings and the average time period spent by tabanids on target surfaces as a function of the surface temperature $T$. For this aim, many different $T$-values of the barrel's surface were necessary. It was all the same that these data are collected by one experimenter observing only the solar sides of the warm and cold barrels, or by two experimenters observing simultaneously both solar and antisolar sides.

We used plastic barrels, the wall (thickness = 5 mm) of which has much lower thermal conductivity than that of metal barrels. Thus, there were significant temperature differences between our warmer and colder barrels during the whole experiment. This temperature difference was further enhanced with the use of iceakkus instead of frozen water (i.e. ice). The thermal capacity of the former is much larger than that of the latter, thus the submerged iceakkus kept the water in our plastic barrels colder for a longer period than submerged ice.

The tabanids landing on the barrels were observed by one (experiments 1–2) or two (experiments 3–5) persons at a distance of 2 m symmetrically from the barrels. These observers wore white clothes and hats against direct sunshine and to minimize their visual attractiveness to tabanids. They were sitting on a chair during the observations. Since we wanted to determine on the spot the species of tabanids landed on our barrels, we observed them with the naked eye 2 m from the barrels, rather than tallying them from a greater distance with binoculars as, for example, Vale [51] performed with tsetse flies. In experiments 1–2, only the solar sides (facing towards the sun) of the barrels were continuously observed by a person (because then only one researcher was available for this task), while in experiments 3–5, both the solar and antisolar (facing towards the antisolar point) sides were observed by two researchers (because then two researchers were available). The direction of observation was perpendicular to the connecting line passing through the centre of the barrels. This connecting line was hourly changed in such a way that its normal vector pointed towards the sun. By this, we ensured that the sunlight hitting the barrels was always nearly parallel to the direction of observation.

During observations, we counted with a 5 min temporal resolution how many landings were on the solar/antisolar sides of the warm/cold barrels, and how many seconds the landed tabanids spent on them. Counting happened with the naked eye, and time periods were measured with a stopwatch. The distribution of the outer surface temperature of the barrels was periodically (experiment 1: half hourly, experiments 2–5: hourly) measured by a thermocamera on their solar and antisolar sides from the direction of observation.

We used dry (non-sticky) barrels, because we wanted to measure the time period spent by alighted tabanids on the barrels. Since our barrels were non-sticky, a given tabanid individual could have landed more than once on them. Such pseudoreplication could be eliminated only if the barrels were sticky. However, from a sticky barrel, a landed tabanid cannot fly away, thus the time period spent by the insect on the barrel cannot be measured. Therefore, we decided not to cover the surface of our barrels with an insect-monitoring glue. This method is frequently used in field experiments studying the attractiveness of certain (e.g. matt) test surfaces to insects (e.g. [15,17,18,52]). Finally, the insect-trapping efficiencies of warm and cold sticky surfaces could be different due to the different viscosities of the warm and cold glue covering the surfaces. This is a further reason why sticky barrels cannot be used in our choice experiments.

## 2.2. Imaging polarimetry

Tabanids are polarotactic insects attracted to (among other cues) linearly polarized light when females and males look for water by horizontal polarization and females seek host animals by high degrees of linear polarization [52]. Therefore, we measured the reflection–polarization characteristics of the two (warm, cold) black barrels used in our field experiments with imaging polarimetry in the red (650 nm), green (550 nm) and blue (450 nm) parts of the spectrum [53,54]. Note, however, that our thermophysiological results on tabanid thermal preference are not influenced by the reflection–polarization patterns of the two optically identical shiny black barrels of different temperatures. The reason for this is the physical fact that the reflection–polarization characteristics of the barrels' surface are independent of its temperature in the visible spectral range. Therefore, all optical parameters (radiance, degree of linear polarization and angle of polarization) of our warm and black barrels were the same. Only tiny water droplets condensed from air humidity on the outer surface of the cold barrel could have resulted in some differences in reflection–polarization, but such a condensation did not occur in our field experiments. The temperature of the antisolar outer surface of the cold barrel was between 20 and 30°C (figure 1), because the plastic barrel's wall with 5 mm thickness was a good thermal insulator. Therefore, the outer surface temperature was never so cold that the air humidity could have been condensed on the wall.

## 2.3. Thermography

The distributions of the surface temperature (thermograms) of two (one sunlit and one shady) white, two brown and two black horses as well as the two black barrels used in our field experiments were measured with an infrared camera (VarioCAM®, Jenoptik Laser Optik Systeme GmbH, Jena, Germany) with a nominal precision of ±1.5°C. Horses were measured under sunny (illuminated with direct sunlight) and shady (when the sun was behind a cloud) conditions. At a given (sunlit or shady) horse, we recorded several thermograms and selected a typical thermogram where the optical axis of the thermocamera was nearly perpendicular to the long axis of the horse. The thermograms of the sunlit and shady sides of barrels were measured periodically (half hourly or hourly) in full sunshine under cloudless sky from the beginning to the end of each experiment. To determine the surface temperature distribution on horses and barrels along straight lines and in arbitrary areas, we used a custom-developed software. On the thermograms of horses, tilted and nearly horizontal lines were selected, along which the average $\langle T \rangle$ and standard deviation $\Delta T$ of the surface temperature $T$ were calculated in order to probe typical $T$-values of sunlit and shady horses. On the barrel's sides, a uniform rectangular area was selected, where $\langle T \rangle$ and $\Delta T$ were determined. The air temperature was not continuously monitored during our experiments, but it changed between about 28 and 36°C from the beginning to the end of the experiments.

The correctness of the temperature measurement of our thermocamera was validated by a calibration procedure with a contact thermometer (GAO Digital Multitester EM392B 06554H, EverFlourish Europe Gmbh, Friedrichsthal, Germany) with a nominal precision of ±1°C. Further details about this calibration can be read in electronic supplementary material, figures S1–S3.

## 2.4. Schlieren imagery

In order to demonstrate the warm/cold air flow in the vicinity of the warm/cold barrels used in our field experiments, we performed Schlieren imagery [55]. With this technique, the spatio-temporal variation of the temperature-dependent refractive index of the air can be visualized. The barrels were modelled by a small

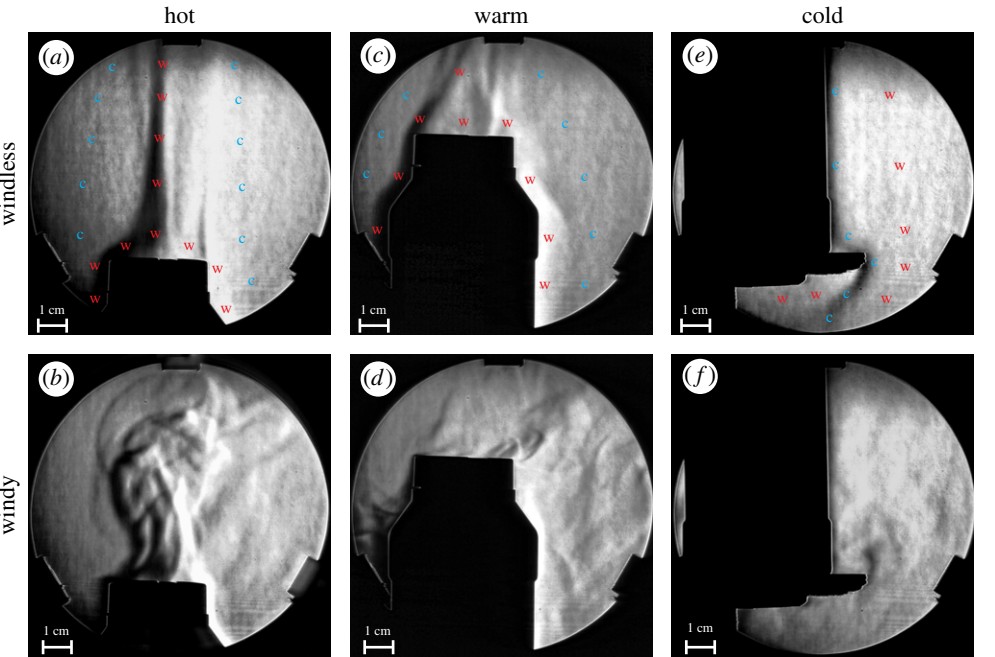

**Figure 3.** Schlieren images of the air next to the hot (*a,b*; *T* = 70℃), warm (*c,d*; *T* = 45℃) and cold (*e,f*; *T* = 8℃) plastic flask modelling the warm/cold barrels used in the choice experiments. In the upper row (*a,c,e*), the air was windless (calm), while in the lower row (*b,d,f*), the air was windy (turbulent), because the flask was very weakly blown by mouth from 1 m. Different shades of grey represent different gradients of the temperature-dependent refractive index of air. w, warmer air; c, colder air.

cylindrical black plastic flask (height = 16 cm, diameter = 7 cm). In its cold state, the flask was filled with tap water and ice cubes, so that the inside water temperature was $T = 0℃$ until the ice melted away. In its warm and hot state, the flask was filled with warm ($T = 47℃$) and hot water ($T = 72℃$), respectively. The surface temperature of the flask was measured with thermography. The cold/warm flask was placed into the light beam of the Schlieren equipment (electronic supplementary material, figures S4 and S5).

## 2.5. Statistical analysis

We applied Wilcoxon signed-rank test (R Statistics 3.2.3) to find differences in tabanid attractiveness between the warm and cold barrels (in experiments 1–2) and between the warm sunlit, warm shady, cold sunlit and cold shady barrel's sides (in experiments 3–5). In the dataset, the number $N$ of landings and the time spent on the barrels were calculated with a 5 min temporal resolution. The statistical analysis compared the same time periods of the solar/antisolar sides of the warm/cold barrels.

# 3. Results

## 3.1. Reflection–polarization and thermal characteristics of the warm and cold barrels

Electronic supplementary material, figure S6 and Results, show that the optical characteristics of the warm and cold barrels used in our field experiments were the same. In figures 1 and 2 and electronic supplementary material, Results, it is clearly seen that in sunshine, the relation among the surface temperatures of the solar and antisolar sides of the warm and cold barrels was: $T_{\text{warm}}(\text{solar}) > T_{\text{warm}}(\text{antisolar}) > T_{\text{cold}}(\text{solar}) > T_{\text{cold}}(\text{antisolar})$. Thus, during the experiments, always four test surfaces (sides of the barrels) with different temperatures and uniform optical characteristics were offered to flying tabanids.

## 3.2. Schlieren imagery of the warm/cold air next to the warm/cold barrels

According to figure 3 and electronic supplementary material, Results, a warm target heats up the boundary layer (i.e. the air next to the flask's wall) and this warm air streams upward due to its lower density than

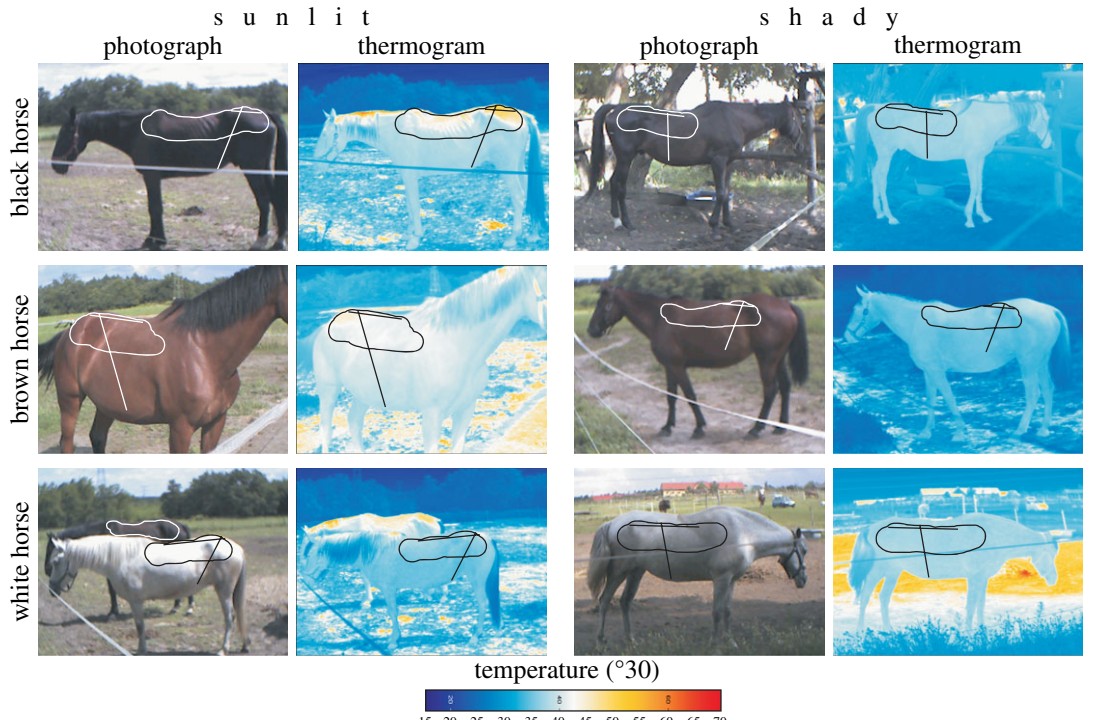

**Figure 4.** Photographs and thermograms of sunlit and shady black, brown and white horses. On the thermograms, the closed curve(s) as well as the tilted and nearly horizontal straight lines represent the regions on/along which the surface temperature was averaged.

that of the surrounding air of 20°C room temperature. On the other hand, a cold target cools down the boundary layer and this cold air streams downward due to its higher density. In calmness (under windless conditions), the thickness of the upward flowing warm and downward streaming cold boundary layer is around 1–2 cm over the surface. Under turbulent (windy) conditions, the warm and cold air layers mix turbulently, thus an elongated warm/cold turbulent train of several centimetres long forms beyond a warm/cold target along the wind direction. Note that tabanids also fly in weaker winds, when the air flow is turbulent around any target (host animals or our barrels).

## 3.3. Surface temperatures of sunlit and shady horses of different colours

Figure 4 shows the thermograms of sunlit and shady white, brown and black horses, and electronic supplementary material, table S1, contains the surface temperatures of a white, a brown and a black horse measured by thermography on an area and along a nearly horizontal and vertical line under sunlit (illuminated by direct sunlight) and shady (sun behind a cloud) conditions. In sunshine, the warmest (44.4–43.1°C) and coldest (36.4°C) were the sunlit body surfaces of the black and white horses, respectively, while the surface temperature of the brown horse was in between (42°C). On the other hand, the mean temperatures of the shady body surfaces of these horses were very similar (35.4–37.3°C). These data demonstrate that in sunshine, the surface temperature $T$ of animals is mainly determined by the albedo $a$ (light absorptance) of the body surface: the lower is $a$, the higher is $T$ [56–59].

## 3.4. Attractiveness of the warm and cold barrels to tabanids

In our field experiment, we observed *T. tergestinus* Egger 1859 tabanid flies. During our experiment, on the horse farm's ground, there was a rectangular black plastic tray (30 × 30 cm) filled with common transparent, yellowish salad oil functioning as an efficient tabanid trap [41]. This trap captured only *T. tergestinus* horseflies. They were identified on the basis of the following morphological characteristics [60, pp. 362–363]: *T. tergestinus* is a medium-sized (length 15–18 mm) species with green eyes and three red bands, the frons in females are narrow, the median callus is linear and connected with the lower callus, the thorax is dark grey, notopleural lobes are dark, the abdomen is reddish-brown at sides from one to four tergites and with dark median stripe with indistinct paler median triangles, while posterior

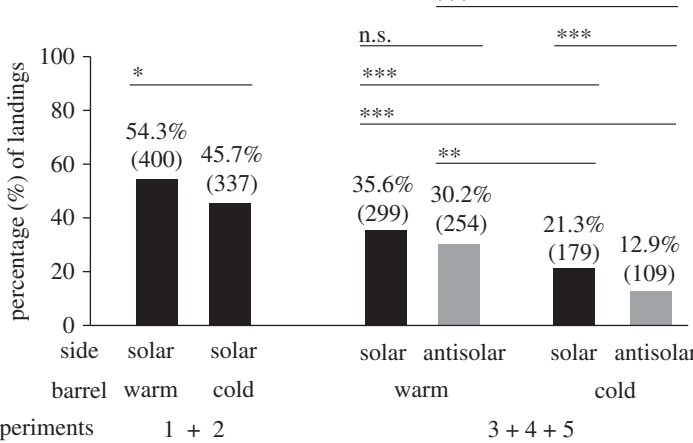

**Figure 5.** Percentage (%) of the total number $N$ (in parentheses) of landings on the solar and antisolar sides of the warm and cold barrels in experiments 1–2 and 3–5 (electronic supplementary material, table S2) with the results of the Wilcoxon signed-rank test. n.s., not significant, $p > 0.05$; *$0.01 < p < 0.05$; **$0.001 < p < 0.01$; ***$p < 0.001$ (electronic supplementary material, table S3).

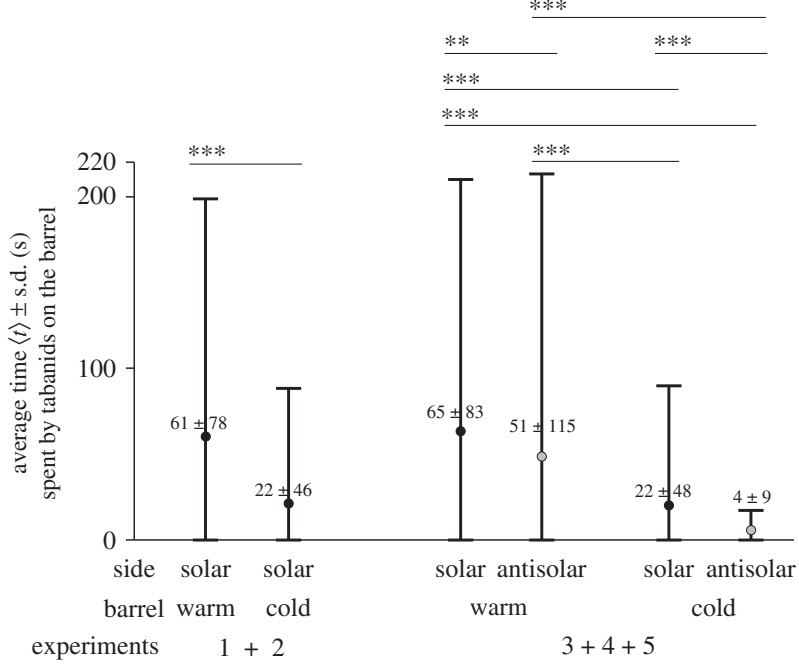

**Figure 6.** Average time $\langle t \rangle \pm$ s.d. (s) per individual spent on the solar and antisolar sides of the warm and cold barrels in experiments 1–2 and 3–5 (electronic supplementary material, table S2) with the results of the Wilcoxon signed-rank test. n.s., not significant, $p > 0.05$; *$0.01 < p < 0.05$; **$0.001 < p < 0.01$; ***$p < 0.001$ (electronic supplementary material, table S3).

tergites are uniformly greyish. Hence, during our experiment, only this tabanid species swarmed in the study area. With the naked eye, we could not detect the sex (male or female) of tabanids landed on the barrels, but they might have been females in all probability, because black targets above the ground level are attacked only by female tabanids [4].

Figures 5 and 6 as well as electronic supplementary material, tables S2 and S3, show the total numbers $N$ of landings on the solar and antisolar sides of the warm and cold barrels used in our field experiments and the average time $\langle t \rangle \pm$ s.d. (second) per individual spent on the different barrel's surface together with the results of the statistical analyses of these data. (i) In experiments 1 and 2, only the solar side of the barrels was observed. On the warm barrel ($N = 400$), significantly ($p = 0.0139$) more tabanids landed than on the cold barrel ($N = 337$). (ii) In experiments 3–5, both the solar and antisolar sides of the barrels were monitored. Then again, the warm barrel ($N = 553$) was highly significantly ($p < 0.0001$) more attractive to tabanids than the cold barrel ($N = 288$). The attractiveness of the different barrel's sides decreased in the following order: $N_{\text{warm}}(\text{solar}) = 299 > N_{\text{warm}}(\text{antisolar}) =$

$254 > N_{cold}$(solar) = 179 > $N_{cold}$(antisolar) = 109 (figure 5). These differences are also significant (electronic supplementary material, table S3A), except the $N_{warm}$(solar) versus $N_{warm}$(antisolar) where $p = 0.0917$. As we have seen above, the mean surface temperature of the barrel's sides decreased also in the same order: $T_{warm}$(solar) > $T_{warm}$(antisolar) > $T_{cold}$(solar) > $T_{cold}$(antisolar) (figures 1 and 2).

Considering the average time $\langle t \rangle$ spent by the alighted individual tabanids on the different barrel's surface, the following trends were obtained (figure 6; electronic supplementary material, tables S2 and S3): (i) in experiments 1–2, the tabanids spent highly significantly ($p < 0.0001$) more time on the solar side of the warm barrel ($\langle t \rangle = 61 \pm 78$ s) than on the solar side of the cold barrel ($22 \pm 46$ s). (ii) In experiments 3–5 again, tabanids spent highly significantly ($p < 0.0001$) more time on the warm barrel ($\langle t \rangle = 58 \pm 99$ s) than on the cold one ($15 \pm 39$ s). (iii) The average time $\langle t \rangle$ spent by tabanids on the different barrel's sides decreased in the following order: $\langle t \rangle_{warm}$(solar) = $65 \pm 83$ s > $\langle t \rangle_{warm}$(antisolar) = $51 \pm 115$ s > $\langle t \rangle_{cold}$(solar) = $22 \pm 48$ s > $\langle t \rangle_{cold}$(antisolar) = $4 \pm 9$ s. These differences are highly significant (electronic supplementary material, table S3B).

Figure 7a shows the total numbers $N$ of landings on the barrels as a function of the barrel's surface temperature $T$. If $T$ was below 18°C ($T < 18$°C), tabanids did not land on the barrels. Although the dispersion (standard deviation) of the $N(T)$ data points is relatively large, the regression line and its 95% confidence area clearly show the tendency that $N$ increases with increasing $T$. Figure 7b represents the average time $\langle t \rangle$ spent by tabanids on the barrels versus $T$ averaged for the periods (30 or 60 min) between two sequential thermographic measurements of $T$. $\langle t \rangle$ increased tendentiously with increasing $T$, and its maximum was 136 s.

# 4. Discussion

The maximum coat temperature of sunlit dark (brown or black) horses (even if they sweat, when the evaporation of sweat decreases the surface temperature) can easily be 50.9–51.5°C (electronic supplementary material, table S1). This is because sunlit surface parts of the dark coat can locally warm up very strongly if the surface is nearly perpendicular to the incident sunlight. Such surface temperatures of sunlit dark horses are fully realistic, because the interior body temperature of adult horses is 37.2–38.3°C (https://thehorse.com/133159/normal-horse-vitals-signs-and-health-indicators/), and the thermal conductivity of horse hair is low. Depending on the orientation of the local body surface relative to direct sunlight, the albedo (low for black stripes, high for white stripes) of the striped coat and the air temperature, the surface temperature of sunlit black zebra stripes, for example, can change between 35 and 55°C [61] or 44 and 56°C [62]. In the top-left (sunlit black horse) and bottom-right (shady white horse) thermograms of figure 4, the temperature of ground with heterogeneous albedo and thermophysical characteristics was approximately 60–65°C. These hot patches correspond to dry soil regions or dung piles, which warmed up very much in direct sunshine.

One of the plausible explanations of the heat preference of tabanids may be based on the fact that an elevated skin temperature of potential hosts is related to either internal (heavy muscular work) or external (strong insolation) heat stresses of the host, the usual thermoregulatory response of which is an increased skin blood flow (and sweating).

## 4.1. A new hypothesis

We propose here the novel hypothesis that the warmer microclimate above the body surface of sunlit dark hosts is a selective advantage for blood-sucking female tabanids: tabanid bites are very painful, which therefore elicit intense defensive behaviour of the host, such as tail flicking, skin twitching, foot stomping and head shaking, for example [63]. Due to these defensive movements, a blood-sucking tabanid can easily get hurt. To avoid such a damage, the biting tabanid should fly away promptly before the host's tail hits it. For such a rapid escape, the tabanid wing muscles should function quickly. Only high enough temperature of these muscles can ensure their quick enough functioning. Thus, we suggest the following new hypothesis:

— Blood-seeking tabanids choose preferentially sunlit darker hosts, partly because their wing muscles and neurons function rapidly enough in the warmer microclimate on the body surface of such hosts in order to escape by flying away from the parasite-repelling reactions of the hosts. (Electrophysiological recordings from flight muscle and associated nerves showed that in certain insects, the number of nerve impulses impinging on a wing muscle is much fewer than that of

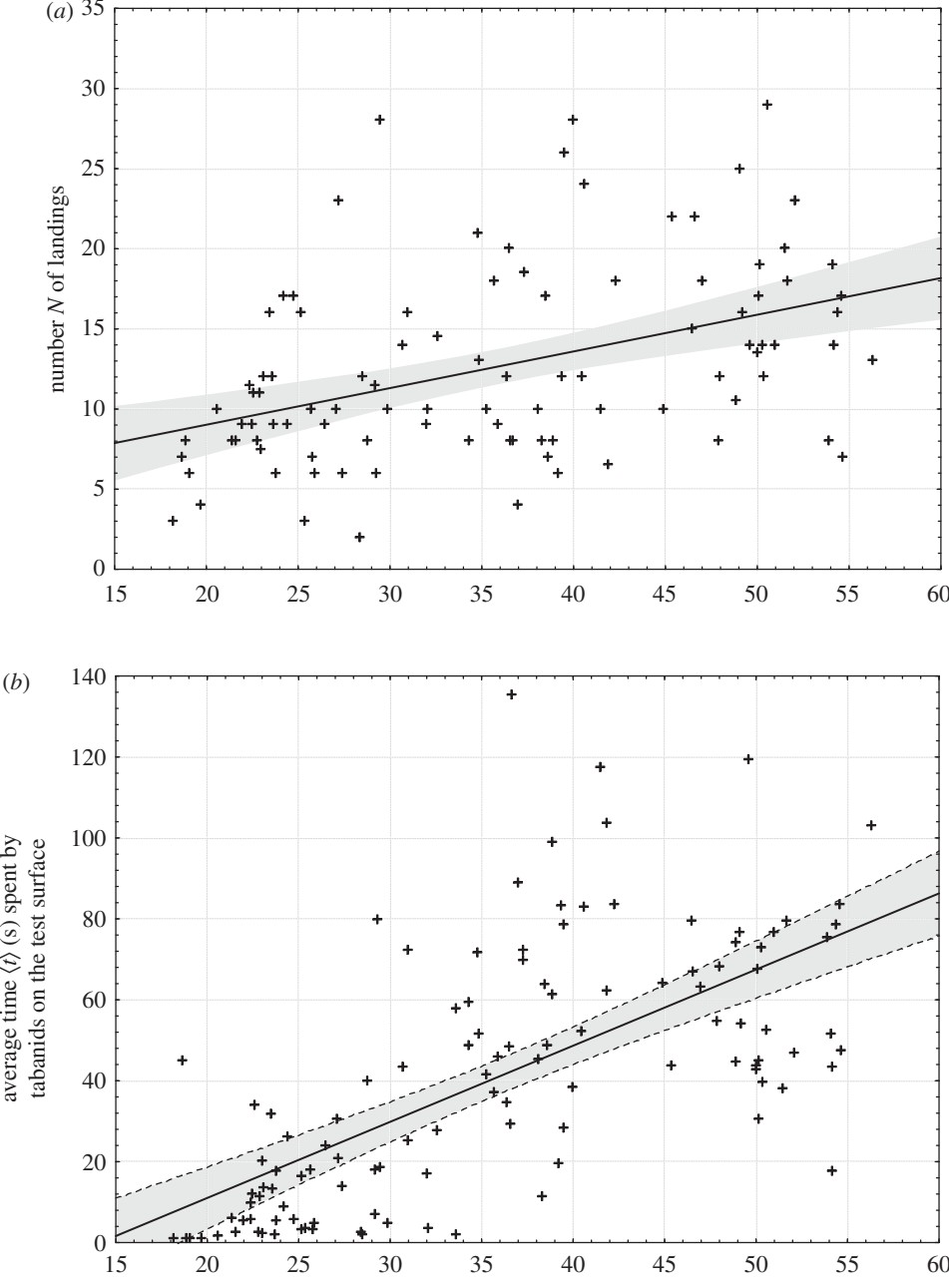

**Figure 7.** (*a*) Total number *N* of landings on the barrels as a function of the barrel's surface temperature *T* (°C). (*b*) Average time $\langle t \rangle = t/N$ (s) spent by tabanids on the barrels versus *T* averaged for the periods (30 or 60 min) between two sequential thermographic measurements of *T*. The straight lines show the regression line fitted to the data points represented by symbol +. The grey area around the regression line shows 95% confidence interval. All temperature data originate from the thermographically measured, temporally changing surface temperatures of the solar and antisolar sides of the warm and cold barrels shown in figure 1.

muscle contractions. In this case, high wingbeat frequencies could not be explained on the basis of a one-to-one relationship between nerve impulse and muscle contraction. On the other hand, in other insects, there is a one-to-one relationship between nerve impulses and muscle contractions [64, p. 218]. In the colder microclimate on shady darker host animals or sunlit/shady brighter hosts, the wing muscles of tabanids may not function rapidly enough for a successful escape.

The prerequisite of this hypothesis is to prove that tabanids indeed prefer warmer hosts against colder ones. In this work, we tested this prerequisite by studying the thermal preference of host-seeking female tabanids.

A possible explanation of the heat preference of tabanids might be related to an association between elevated skin temperature and increased skin blood flow of a potential host animal. However, it is also possible that the clearly observed warmth preference is an indication of the thermal strategy of tabanids [65], where they prefer environments that keep an optimal body temperature. When a female tabanid lands on a host animal, her wing muscles are still sufficiently warm for flying, because prior to alighting, the muscles were kept warm by the heat produced by muscle contractions. According to our hypothesis, during flight, she may sense with her thermoreceptors placed on her body surface that the boundary air layer of the picked host is warm enough. For this rapid thermoreception, the short time period could be sufficient during which she overflies through the host's boundary layer.

In our opinion, both the warm/cold boundary layer of a few centimetres and the warm/cold turbulent air train of a few decimetres next to the target's wall (figure 3) are important to explain the behaviour of tabanids flying around the barrels in our field experiments: within a few to several centimetres, flying tabanids may sense the temperature of the boundary layer (microclimate) and thus could estimate the temperature of the barrel's surface and decide whether they do or do not alight on it.

On the basis of the possibly promptly sensed temperature of the boundary layer, a female tabanid could decide whether the host's body surface is or is not warm enough for alighting. When she lands on the host, her non-contracting wing muscles begin to cool, the speed of which depends on the temperature of the boundary layer over hosts as follows:

— In a warm boundary layer (sunlit dark host), the cooling process of wing muscles is slow, or it is even negligible, thus a blood-sucking female tabanid can fly away quickly when the host wants to swat her (by tail, for example). According to our new hypothesis, this can be one of the reasons why tabanids prefer sunlit dark host animals.
— In a cooler boundary layer (shady dark or sunlit bright host), the cooling process of wing muscles is rapid, thus a blood-sucking tabanid cannot fly away quickly when the host wants to swat her. In our opinion, this can be one of the reasons why tabanids dislike bright host animals and shady dark hosts.

When a female tabanid is seeking the veins directly below the skin of the host, she does not immediately bite the host. This seeking lasts a minimum of a few seconds, but it is generally much longer. During this time, the temperature of her wing muscles can reach that of the host's boundary layer. According to our hypothesis, if this temperature is high enough (sunlit dark host), then after the start of blood-sucking with a painful bite, the tabanid is able to avoid the host's parasite-repelling reactions by a rapid flying away, which reactions are very dangerous for a blood-sucking fly. If the temperature of the boundary layer is not high enough, then the wing muscles cooled down in it are not able to function quickly enough; therefore, after the start of blood-sucking, the tabanid is not able to avoid the host's parasite-repelling reactions by flying away.

It is an established fact that muscles of ectothermic animals function less well and contract less rapidly and less forcefully under cold conditions, and the temperature differences measured by us in this work between horses in the sun versus shade, or black versus white horses, are physiologically clearly relevant. Our idea to link the activity and readiness of the flight musculature with ambient temperature is therefore justified.

It has been shown for numerous ectothermic animals from earthworms [66] to Antarctic fish [67] that neuronal function is strongly dependent on environmental temperature, so that conduction velocities are linearly related to temperature. Not only are conduction velocities faster at higher temperatures, synaptic transmission (together with conduction crucial for initiating a muscular contraction) would also be faster at higher temperature. Since poikilothermic (cold-blooded) animals can react more swiftly to the approach of a threat, be this a swishing tail or another animal, under a warm condition, tabanids do not need to depart too quickly, but can rely on their rapid reaction to take off faster (when necessary) than if they were colder. This extra advantage of being able to stay a little longer on the host when warm allows the tabanid to ingest a little more blood. And this could be one additional (if not the real reason) for tabanids to seek out warmer hosts.

Feeding on a warmer host may have the advantage that tabanids could feed faster (e.g. because of the faster functioning of the musculature of their mouthparts [32]) with which they could minimize the time spent on the host, and thus minimize the hosts' anti-parasitic behaviours [4]. However, this has the potential cost of thermal stress [65]. Another advantage could be that feeding faster minimizes the decrease in the temperature of the insect's body, including wing muscles [64]. Note that both advantages are also related to the faster functioning of tabanid muscles.

## 4.2. Fulfilment of the prerequisite of the new hypothesis

The prerequisite of the above hypothesis is that tabanids prefer warmer hosts against colder ones. The results of our field experiments show that this prerequisite is fulfilled in the case of *T. tergestinus* which horsefly species evidently preferred the warmer black barrel against the colder one, and both targets possessed the same optical characteristics. Since a remotely thermosensitive organ detecting infrared radiation by tabanids has not been discovered, the only explanation of our finding is that prior to alighting, the tabanids flying close around the barrels sense the colder air (boundary) layer next to the vertical wall of the colder barrel. This boundary layer was not warm enough, thus tabanids landed only rarely on the colder barrel, and if they looked for blood, they remained only for a very short periods on the cold substratum.

Although we could not determine the sex of tabanids (*T. tergestinus*) in our field experiments, most of them were females in all probability, because such targets attract only females. On the one hand, it is well known that the shiny black sphere of the Manitoba trap attracts exclusively female tabanids; thus, this and similar traps capture only females [4,8,9,10,19,21,22,23,27,38,68]. On the other hand, in our earlier field experiments, the shiny black sticky or dry (non-sticky) vertical traps caught only female tabanids [12,15,16,33,35,40,48,49]. Therefore, our results are valid for female *T. tergestinus* horseflies.

Finally, our hypothesis is also supported by the fact that there is an optimal interval of the air temperature for the flight (muscles) of tabanids. This has been found by Herczeg *et al*. [49], for example: the six day-active tabanid species (*Atylotus loewianus* Villeneuve 1920, *T. tergestinus* Egger 1859, *Tabanus bovinus* Linnaeus 1758, *Tabanus maculicornis* Zetterstedt 1842, *Tabanus bromius* Linnaeus 1758, *Haematopota pluvialis* Linnaeus 1758) studied by Herczeg *et al*. [49] need an ambient air temperature of at least 18°C to fly. This temperature threshold corresponds to the minimum temperature required to activate the enzyme function and flight muscle fibres of tabanids [69]. Herczeg *et al*. [49] found that 31–35°C is the optimal temperature range for the flight of the mentioned tabanids. This result confirms the similar findings of Amano [70]. Kohane & Watt [71] and Josephson *et al*. [72] have observed that the optimal frequency and power of isolated flight muscles increase as the temperature rises, peaking at about 39°C.

As far as we know, quantitative measurements of the rate of heat transfer from the wing muscles of flies to the ambient air have not been performed. Thus, there is no information about the relevant time periods during which the wing muscles of tabanids can warm up from the local environmental (air or substratum) temperature to the minimum temperature necessary for efficient flight. Only general knowledge is available in this subject: tabanids (as other flies) are ectothermic, that is, their body temperature changes with the environment [32]. As flying insects in general, tabanids have to maintain their flight muscles above a minimum temperature to gain power enough to fly. Since smaller tabanids (as animals in general) have larger surface per volume ratio, smaller tabanids cool down more quickly than larger ones. Flying tabanids (as other insects) raise their body temperature through the action of flight, above environmental temperatures [73]. Vibrating or shivering the wing muscles and basking, for example, allow tabanids (as other insects) to actively increase the temperature of their flight muscles, enabling flight [73]. From this point of view, warmth preference is a trivial thermal strategy to minimize active energy consumption for maintaining an optimal body temperature.

## 4.3. A previous study on the thermal preference of tabanids

Using two shiny black and two shiny white metal barrels filled with warm or cold water as decoys in a pair of Manitoba canopy traps, Bracken & Thorsteinson [28] studied the attractiveness to tabanids of thermally different targets. Their canopy traps with a warmer (31.1°C) and a colder (11°C) black barrel captured $308 \pm 60.5$ and $227 \pm 39$ tabanids (species not published), while those with a warmer (34.6°C) and a colder (22°C) white barrel caught $17 \pm 8$ and $9 \pm 3$ tabanids, respectively. They concluded that decoys with surface temperature approximating to body temperature of mammals are not significantly more attractive to tabanids than decoys at air temperature. Our results and conclusions (see later) obtained for *T. tergestinus* partly contradict those of Bracken & Thorsteinson [28]. The reason for this is that we used a different method. Bracken & Thorsteinson [28] captured tabanids with Manitoba canopy traps having black or white decoys with different temperatures. This trap type cannot capture all tabanids landed on the decoy [68]. While Bracken & Thorsteinson [28] determined the attractiveness to tabanids only for two surface temperatures (31.1°C and 11°C) of their black decoys, measuring at 98 different temperatures, we determined the thermal preference of tabanids as a function of the target's surface temperature.

# 5. Conclusion

From our results, we conclude the following:

(1) The warmer the shiny black barrel's surface, the more *T. tergestinus* tabanid flies alighted on it (figures 5 and 7*a*). In spite of the fact that the optical characteristics of the warm and cold barrels were the same, significantly more tabanids landed on the warm barrel than on the cold barrel. Hence, these tabanids prefer warmer dark targets with uniform optical characteristics against colder ones.

(2) From conclusion (1), it follows that tabanids decided to land on the barrels not only by means of optical cues, but also thermal cues governed their choice.

(3) We have frequently observed that tabanids flew so close to the barrel's surface that their wings almost touched it, and often, we also heard a typical knock sound produced by their body when they hit the barrel during flight. From this observation and on the basis of the Schlieren measurements and (1) and (2), we conclude that tabanids flying in the immediate vicinity of the barrels can sense the temperature of the air (boundary) layer next (a few centimetres in calmness and a few decimetres in wind) to the barrel's surface, and the warmer is it, the higher is the chance of their landing.

(4) The warmer was the barrel's surface, tabanids spent the longer time on it (figures 6 and 7*b*).

All these support our main conclusion that *T. tergestinus* tabanid horseflies prefer warmer alighting surfaces (host animals) against colder ones with the same optical properties (remote visual attractiveness). This finding corroborates our hypothesis that biting tabanids should fly away promptly before the host's tail hits them and that for such rapid escapes, the tabanid wing muscles should function optimally, i.e. with as little delay as possible. We suggest that biting tabanids seek out warmer hosts, because only sufficiently high temperatures can ensure the rapid and good functioning of their flight muscles and neurons.

Data accessibility. Our paper has the following electronic supporting materials: Supplementary Materials and methods, Calibration of the thermocamera, Supplementary results, Reflection–polarization characteristics of the warm and cold barrels, Thermal characteristics of the warm and cold barrels, Schlieren imagery of the warm/cold air next to the warm/cold barrels, Attractiveness of the warm and cold barrels to tabanids; electronic supplementary material, tables S1–S3 and figures S1–S7.

Authors' contributions. G.H. and Á.P.: substantial contributions to conception and design; G.H., Á.P., S.P., T.T. and I.M.J.: performing experiments and data acquisition; G.H., Á.P., S.P. and T.T.: data analysis and interpretation; G.H., Á.P. and I.M.J.: drafting the article or revising it critically for important intellectual content.

Competing interests. The authors have no competing interests.

Funding. This work was supported by the grant no. NKFIH K-123930 (Experimental Study of the Functions of Zebra Stripes: A New Thermophysiological Explanation) received by G.H. from the Hungarian National Research, Development and Innovation Office.

Acknowledgements. We are grateful to István Simon, who permitted the performance of our field experiment and thermographic measurements in his Szálender horse farm in Göd, Hungary. We thank the logistic and technical help of Dr András Barta and Péter Takács (Estrato Research and Development Ltd). We thank Prof. Tim Caro (Department of Wildlife, Fish and Conservation Biology, University of California, Davis, CA, USA) for reading and commenting an earlier version of our manuscript. We are grateful to two anonymous reviewers for their constructive and valuable comments.

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
