## [Reviewer comments · Royal Society Open Science]

Review History

RSOS-191119.R0 (Original submission)

Review form: Reviewer 1

Is the manuscript scientifically sound in its present form?

Yes

Are the interpretations and conclusions justified by the results?

Yes

Is the language acceptable?

Yes

Do you have any ethical concerns with this paper?

No

Have you any concerns about statistical analyses in this paper?

No

Recommendation?

Accept with minor revision (please list in comments)

Comments to the Author(s)

Reviewer's comments

July 10, 2019

I suggest some corrections mainly related for identification of species *Tabanus tergustinus*.

My comments about the manuscript are as follows:

Page 3, line 19: you should insert the following reference (Perich et al. 1986) after the sentence 'Cattle annoyed by tabanids grow thinner, thus their meat and milk production decreases.'

Perich M. J., Wright R. E., Lusby K. S. (1986) Impact of horse flies (Diptera: Tabanidae) on beef cattle. *Journal of Economic Entomology* 79: 128-131.

Page 3, line 59: you should insert the following reference (Krčmar et al. 2014), because of following reasons: Recently, the efficiency of ten differently colored modified box traps for collecting tabanids was studied in eastern Croatia. In this study the black modified trap was the most successful and collected 20% of all collected tabanids.

Krčmar S., Radolić V., Lajoš P., Lukačević I. (2014) Efficiency of colored modified box traps for sampling tabanids. *Parasite* 21: 67 (doi: 10.1051/parasite/2014068)

Page 6, line 5: Identification of horseflies on the experimental plot (barrels) in field experiment with the naked eye on distance of 2 m from the experimental plot (barrels) are very difficult, almost impossible! *Tabanus tergustinus* is the medium sized horse flies with length from 15 to 18 mm, from distance of 2 m red stripes on green eyes are not visible as stated on the page 8, line 25. Very similar species to *T. tergustinus* on the first view is *Hybomitra ciureai*, both species have green eyes with three stripes and reddish-brown sidemarkings on tergites from 1 to 4. The probability of wrong identification is great. I suppose the authors have sampled and identified several species of *Tabanus tergustinus* by key (Chvala et al. 1972, p. 362-363) before the beginning of the experiment and that on that basis they followed the landing of horseflies on experimental plot (barrels). Following morphological characteristics important to identification of species *Tabanus tergustinus* should be included in the manuscript. *Tabanus tergustinus* is the medium sized species with eyes naked and three banded, frons in female narrow, median callus linear and connected with lower callus, thorax is dark grey, notopleural lobes dark, abdomen is reddish-brown at sides from one to four tergites and with dark median stripe with indistinct paler median triangles, while posterior tergites uniformly greyish (Chvála et al. 1972).

Chvála M., Lyneborg L., Moucha J. (1972) The horse flies of Europe (Diptera, Tabanidae). Entomological Society of Copenhagen. Copenhagen, Denmark.

Page 10, line 16 to 20: At the end of the sentence which begins 'We suggest... (16 line to 20 line) you should insert the following: However, electrophysiological recordings from flight muscle and associated nerves showed that in certain insects the number of nervous impulses impinging on a wing muscle are much fewer than the number of muscle contractions. Thus, the high wingbeat frequency could not be explained on the basis of a one to one relationship between nervous impulse and muscle contraction. While in other cases there was a one to one relationship between nerve impulses and muscle contraction (Romoser and Stoffolano 1998, p. 218).

Romoser W. S., Stoffolano J. G. Jr. (1998) The science of entomology. WCB McGraw-Hill Companies, Inc. USA.

Review form: Reviewer 2

Is the manuscript scientifically sound in its present form?

No

Are the interpretations and conclusions justified by the results?

No

Is the language acceptable?

Yes

Do you have any ethical concerns with this paper?

No

Have you any concerns about statistical analyses in this paper?

Yes

Recommendation?

Major revision is needed (please make suggestions in comments)

Comments to the Author(s)

In this paper, Horváth et al. investigated the attraction of tabanids to objects at different temperatures. The authors quantified the insects' landing behavior and time spent on barrels at different temperatures while taking into account the optical properties of the objects. The authors show that tabanids are attracted to warmer objects and spend more time resting on these compared to a cooler object. Overall, I think that the topic of the study is interesting and brings some knowledge to the field of medical and veterinary entomology. The paper is overall well written despite of a very long discussion with unclear and unnecessary justifications. The introduction is rich in references. The objectives are clearly defined, and figures are clear. However, this paper has some major issues related to controls and despite a good coverage of the literature available of tabanids, some important notions and concepts related to host-seeking behavior in blood-sucking insects are missing. Specific comments can be found below:

Introduction

- Remove the page number for the references to Lehane's book (e.g. P2 L8)
- The introduction is rich in references which is important, but some details might be removed so the main message is not diluted (e.g. P3 L2-19)
- I am surprised to see only 1 reference to work conducted on tsetse flies despite the huge amount of literature available on host preference, host seeking (including thermopreference and visual cues) and their similarities in feeding behavior.
- P3 L24: a host, if sweating more, will have a bacteria population growing differently compared to a less sweaty host, which can consequently affect chemicals released by the host's skin.

M&M

- P4 L36: It is unclear whether the warm barrel was filled with water or not. Water vapor is another important cue for host-seeking and if water vapor is emanating from one the barrel and not this other, that could affect the results.

- I am concerned about the humidity on the cold barrel due to condensation that could affect the vision of the fly.
- Did you monitor the ambient temperature during your experiments?
- P4 L48-L54: I would remove the results from exp. 1 from the analysis then.
- P5: Please describe more the behavior of the experimenters during the observations? Were they sitting or walking around the barrels?
- P5: Based on the issues that arose during experiment 1 and the fact that the number of observers was different between the experiments, I strongly recommend removing data for exp 1 and 2 for the final analysis. This will make the results cleaner and more replicable.
- P5 L17-22: how did you count the number of landing? Did you use a counter? Why not using video recordings? That would have allowed you to avoid any bias associated with the experimenters being around the tank, including the fact that they released CO₂ and other body odors, humidity and heat (all cues used by these flies to locate a host).
- P5 L50-54: I have a hard time imagining that a barrel containing water at 0°C would not create condensation at ambient temperature. What was the temperature of the environment during the experiments? What about relative humidity?
- P5/P6: thermography with horses: how many measurements did you do? How many horses did you use for your measurements? What value was the emissivity set up at?
- P6: Why testing turbulent / windy conditions if tabanids avoid flying under such conditions?

Results

- Figure 4: the scale is too large to notice any difference in temperature across the body of the horses. Also, related to P8 L60: the surrounding area (unless you changed the emissivity for these specific ROIs) might not be warmer or cooler, depending on the specific emissivity of the material.
- Figure 7: Is this data related to the warm barrel cooling down or the cool barrel warming up (or both)? Any stats / regression for this data?
- Table S3: did you use a statistical correction to adjust for multiple comparisons?
- Why so many flies landed on the cold barrel? You mentioned that no flies landed on the barrel when T > 18°C. It is unclear to me how the temperature of the barrel changed over time and why it was not maintained at one given T for the whole tests. If T changed over time, then it is affecting your statistical analyses unless you first create categories / ranges and then compare them.

Discussion

- The discussion is centered on the hypothesis that the authors make based on their results. It would be interesting to discuss the results in the context of other papers and to expand to other biological systems including tsetse flies. Also, key literature on insects flying at very low temperature including work by B. Heinrich should be included and discussed.
- The discussion is also ignoring our current knowledge on host-seeking behavior in blood-sucking insects and the sequence of events regarding the use long-range and short-range cues. I suggest digging into this literature as well.
- P8: The first part of the discussion is useless (L4-45) and should be removed along with data from experiments 1 and 2 as stated before. That would make this paper and its take home message stronger. Moreover, some of these are overstatements or just impossible to verify (e.g. the CO₂ released by the two observers was similar and did not affect the experiments).
- P9: The flight muscles of these insects probably don't really cool down once they are on the host because of 1) their ability to feed extremely fast which help them stay warm and 2) their ability to maintain these muscles by shivering. You would need thermographic analyses to support your hypothesis regarding the cooling of these muscles for example or thermocouple recordings.
- Feeding on a warmer host presents the advantage of being able to feed faster and thus minimizing the time spent of the host and consequently minimizing the hosts' anti-parasitic behaviors. This has a potential cost too: thermal stress. Another advantage is that feeding faster

minimizes the decrease in body temperature of the insect. This should be discussed and included here.

Decision letter (RSOS-191119.R0)

31-Jul-2019

Dear Dr Horvath,

The editors assigned to your paper ("Attractiveness of thermally different, uniformly black targets to horseflies: *Tabanus tergustinus* prefers sunlit warm shiny dark targets") have now received comments from reviewers. We would like you to revise your paper in accordance with the referee and Associate Editor suggestions which can be found below (not including confidential reports to the Editor). Please note this decision does not guarantee eventual acceptance.

Please submit a copy of your revised paper before 23-Aug-2019. Please note that the revision deadline will expire at 00.00am on this date. If we do not hear from you within this time then it will be assumed that the paper has been withdrawn. In exceptional circumstances, extensions may be possible if agreed with the Editorial Office in advance. We do not allow multiple rounds of revision so we urge you to make every effort to fully address all of the comments at this stage. If deemed necessary by the Editors, your manuscript will be sent back to one or more of the original reviewers for assessment. If the original reviewers are not available, we may invite new reviewers.

- Data accessibility

It is a condition of publication that all supporting data are made available either as supplementary information or preferably in a suitable permanent repository. The data accessibility section should state where the article's supporting data can be accessed. This section

should also include details, where possible of where to access other relevant research materials such as statistical tools, protocols, software etc can be accessed. If the data have been deposited in an external repository this section should list the database, accession number and link to the DOI for all data from the article that have been made publicly available. Data sets that have been deposited in an external repository and have a DOI should also be appropriately cited in the manuscript and included in the reference list.

If you wish to submit your supporting data or code to Dryad (<http://datadryad.org/>), or modify your current submission to dryad, please use the following link:
<http://datadryad.org/submit?journalID=RSOS&manu=RSOS-191119>

- **Competing interests**

- **Authors' contributions**

- **Acknowledgements**

- **Funding statement**

Kind regards,

Alice Power

Editorial Coordinator

on behalf of Dr Jake Socha (Associate Editor) and Kevin Padian (Subject Editor)
openscience@royalsociety.org

Associate Editor's comments (Dr Jake Socha):

The reviewers agreed that this manuscript presenting interesting and novel work. However, there remain major issues that will need to be addressed in order to be considered for publication. In addition to addressing the technical questions about the experiments and data analysis, there was concern that the introduction was overly long and lacked focus, the materials and methods lack critical details on data collection and analysis, and the discussion should incorporate knowledge of the thermal biology of blood-sucking insects. Please address the reviewers' concerns carefully in the revised submission.

Comments to Author:

Reviewers' Comments to Author:

Reviewer: 1

Comments to the Author(s)

Reviewer's comments

July 10, 2019

I suggest some corrections mainly related for identification of species *Tabanus tergustinus*.

My comments about the manuscript are as follows:

Page 3, line 19: you should insert the following reference (Perich et al. 1986) after the sentence 'Cattle annoyed by tabanids grow thinner, thus their meat and milk production decreases.'

Perich M. J., Wright R. E., Lusby K. S. (1986) Impact of horse flies (Diptera: Tabanidae) on beef cattle. *Journal of Economic Entomology* 79: 128-131.

Page 3, line 59: you should insert the following reference (Krčmar et al. 2014), because of following reasons: Recently, the efficiency of ten differently colored modified box traps for collecting tabanids was studied in eastern Croatia. In this study the black modified trap was the most successful and collected 20% of all collected tabanids.

Krčmar S., Radolić V., Lajoš P., Lukačević I. (2014) Efficiency of colored modified box traps for sampling tabanids. *Parasite* 21: 67 (doi: 10.1051/parasite/2014068)

Page 6, line 5: Identification of horseflies on the experimental plot (barrels) in field experiment with the naked eye on distance of 2 m from the experimental plot (barrels) are very difficult, almost impossible! *Tabanus tergustinus* is the medium sized horse flies with length from 15 to 18 mm, from distance of 2 m red stripes on green eyes are not visible as stated on the page 8, line 25. Very similar species to *T. tergustinus* on the first view is *Hybomitra ciureai*, both species have green eyes with three stripes and reddish-brown sidemarkings on tergites from 1 to 4. The probability of wrong identification is great. I suppose the authors have sampled and identified several species of *Tabanus tergustinus* by key (Chvala et al. 1972, p. 362-363) before the beginning of the experiment and that on that basis they followed the landing of horseflies on experimental plot (barrels). Following morphological characteristics important to identification of species *Tabanus tergustinus* should be included in the manuscript. *Tabanus tergustinus* is the medium sized species with eyes naked and three banded, frons in female narrow, median callus linear and connected with lower callus, thorax is dark grey, notopleural lobes dark, abdomen is reddish-brown at sides from one to four tergites and with dark median stripe with indistinct paler median triangles, while posterior tergites uniformly greyish (Chvála et al. 1972).

Chvála M., Lyneborg L., Moucha J. (1972) The horse flies of Europe (Diptera, Tabanidae). Entomological Society of Copenhagen. Copenhagen, Denmark.

Page 10, line 16 to 20: At the end of the sentence which begins 'We suggest... (16 line to 20 line) you should insert the following: However, electrophysiological recordings from flight muscle and associated nerves showed that in certain insects the number of nervous impulses impinging on a wing muscle are much fewer than the number of muscle contractions. Thus, the high wingbeat frequency could not be explained on the basis of a one to one relationship between nervous impulse and muscle contraction. While in other cases there was a one to one relationship between nerve impulses and muscle contraction (Romoser and Stoffolano 1998, p. 218).

Romoser W. S., Stoffolano J. G. Jr. (1998) The science of entomology. WCB McGraw-Hill Companies, Inc. USA.

Reviewer: 2

Comments to the Author(s)

In this paper, Horváth et al. investigated the attraction of tabanids to objects at different temperatures. The authors quantified the insects' landing behavior and time spent on barrels at different temperatures while taking into account the optical properties of the objects. The authors show that tabanids are attracted to warmer objects and spend more time resting on these compared to a cooler object. Overall, I think that the topic of the study is interesting and brings some knowledge to the field of medical and veterinary entomology. The paper is overall well written despite of a very long discussion with unclear and unnecessary justifications. The introduction is rich in references. The objectives are clearly defined, and figures are clear. However, this paper has some major issues related to controls and despite a good coverage of the literature available of tabanids, some important notions and concepts related to host-seeking behavior in blood-sucking insects are missing. Specific comments can be found below:

Introduction

- Remove the page number for the references to Lehane's book (e.g. P2 L8)
- The introduction is rich in references which is important, but some details might be removed so the main message is not diluted (e.g. P3 L2-19)
- I am surprised to see only 1 reference to work conducted on tsetse flies despite the huge amount of literature available on host preference, host seeking (including thermopreference and visual cues) and their similarities in feeding behavior.
- P3 L24: a host, if sweating more, will have a bacteria population growing differently compared to a less sweaty host, which can consequently affect chemicals released by the host's skin.

M&M

- P4 L36: It is unclear whether the warm barrel was filled with water or not. Water vapor is another important cue for host-seeking and if water vapor is emanating from one the barrel and not this other, that could affect the results.
- I am concerned about the humidity on the cold barrel due to condensation that could affect the vision of the fly.
- Did you monitor the ambient temperature during your experiments?
- P4 L48-L54: I would remove the results from exp. 1 from the analysis then.
- P5: Please describe more the behavior of the experimenters during the observations? Were they sitting or walking around the barrels?

- P5: Based on the issues that arose during experiment 1 and the fact that the number of observers was different between the experiments, I strongly recommend removing data for exp 1 and 2 for the final analysis. This will make the results cleaner and more replicable.
- P5 L17-22: how did you count the number of landing? Did you use a counter? Why not using video recordings? That would have allowed you to avoid any bias associated with the experimenters being around the tank, including the fact that they released CO₂ and other body odors, humidity and heat (all cues used by these flies to locate a host).
- P5 L50-54: I have a hard time imagining that a barrel containing water at 0°C would not create condensation at ambient temperature. What was the temperature of the environment during the experiments? What about relative humidity?
- P5/P6: thermography with horses: how many measurements did you do? How many horses did you use for your measurements? What value was the emissivity set up at?
- P6: Why testing turbulent / windy conditions if tabanids avoid flying under such conditions?

Results

- Figure 4: the scale is too large to notice any difference in temperature across the body of the horses. Also, related to P8 L60: the surrounding area (unless you changed the emissivity for these specific ROIs) might not be warmer or cooler, depending on the specific emissivity of the material.
- Figure 7: Is this data related to the warm barrel cooling down or the cool barrel warming up (or both)? Any stats / regression for this data?
- Table S3: did you use a statistical correction to adjust for multiple comparisons?
- Why so many flies landed on the cold barrel? You mentioned that no flies landed on the barrel when $T > 18^{\circ}\text{C}$. It is unclear to me how the temperature of the barrel changed over time and why it was not maintained at one given T for the whole tests. If T changed over time, then it is affecting your statistical analyses unless you first create categories / ranges and then compare them.

Discussion

- The discussion is centered on the hypothesis that the authors make based on their results. It would be interesting to discuss the results in the context of other papers and to expand to other biological systems including tsetse flies. Also, key literature on insects flying at very low temperature including work by B. Heinrich should be included and discussed.
- The discussion is also ignoring our current knowledge on host-seeking behavior in blood-sucking insects and the sequence of events regarding the use long-range and short-range cues. I suggest digging into this literature as well.
- P8: The first part of the discussion is useless (L4-45) and should be removed along with data from experiments 1 and 2 as stated before. That would make this paper and its take home message stronger. Moreover, some of these are overstatements or just impossible to verify (e.g. the CO₂ released by the two observers was similar and did not affect the experiments).
- P9: The flight muscles of these insects probably don't really cool down once they are on the host because of 1) their ability to feed extremely fast which help them stay warm and 2) their ability to maintain these muscles by shivering. You would need thermographic analyses to support your hypothesis regarding the cooling of these muscles for example or thermocouple recordings.
- Feeding on a warmer host presents the advantage of being able to feed faster and thus minimizing the time spent of the host and consequently minimizing the hosts' anti-parasitic behaviors. This has a potential cost too: thermal stress. Another advantage is that feeding faster minimizes the decrease in body temperature of the insect. This should be discussed and included here.

Author's Response to Decision Letter for (RSOS-191119.R0)

See Appendix A.

RSOS-191119.R1 (Revision)

Review form: Reviewer 1

Is the manuscript scientifically sound in its present form?

Yes

Are the interpretations and conclusions justified by the results?

Yes

Is the language acceptable?

Yes

Do you have any ethical concerns with this paper?

No

Have you any concerns about statistical analyses in this paper?

No

Recommendation?

Accept as is

Comments to the Author(s)

Manuscript is acceptable in present form.

Review form: Reviewer 2

Is the manuscript scientifically sound in its present form?

Yes

Are the interpretations and conclusions justified by the results?

Yes

Is the language acceptable?

Yes

Do you have any ethical concerns with this paper?

No

Have you any concerns about statistical analyses in this paper?

No

Recommendation?

Accept with minor revision (please list in comments)

Comments to the Author(s)

I thank the authors for their thorough revision of the manuscript.

- I suggest adding some references regarding the trade-off between thermal stress and time spent of the host (subsection 4.1).

- I would also add that using video recordings of landing behavior in the field is absolutely feasible and can be pretty cheap if using simple cameras such as GoPro.

- Finally, several thermographic systems (e.g. FLIR) allow experimenters to calibrate the emissivity during recording and post-recording by ROIs. This has nothing to do with other internal calibration that cameras do during recordings. Depending on the camera you used, your system might not correct for this automatically or might not allow for emissivity adjustments (fixed emissivity): <https://www.jenoptik.us/products/cameras-and-imaging-modules/thermography-camera/handheld-thermographic-camera>. In any case, it seems essential to control for this value and to add it to the material and method section. FYI, by default, the emissivity is often set up at 0.98.

Decision letter (RSOS-191119.R1)

08-Sep-2019

Dear Dr Horvath:

On behalf of the Editors, I am pleased to inform you that your Manuscript RSOS-191119.R1 entitled "Attractiveness of thermally different, uniformly black targets to horseflies: *Tabanus tergustinus* prefers sunlit warm shiny dark targets" has been accepted for publication in Royal Society Open Science subject to minor revision in accordance with the referee suggestions. Please find the referees' comments at the end of this email.

The reviewers and Subject Editor have recommended publication, but also suggest some minor revisions to your manuscript. Therefore, I invite you to respond to the comments and revise your manuscript.

- Ethics statement

- Data accessibility

It is a condition of publication that all supporting data are made available either as supplementary information or preferably in a suitable permanent repository. The data accessibility section should state where the article's supporting data can be accessed. This section should also include details, where possible of where to access other relevant research materials such as statistical tools, protocols, software etc can be accessed. If the data has been deposited in an external repository this section should list the database, accession number and link to the DOI for all data from the article that has been made publicly available. Data sets that have been

deposited in an external repository and have a DOI should also be appropriately cited in the manuscript and included in the reference list.

If you wish to submit your supporting data or code to Dryad (<http://datadryad.org/>), or modify your current submission to dryad, please use the following link:
<http://datadryad.org/submit?journalID=RSOS&manu=RSOS-191119.R1>

- **Competing interests**

- **Authors' contributions**

- **Acknowledgements**

- **Funding statement**

Because the schedule for publication is very tight, it is a condition of publication that you submit the revised version of your manuscript before 17-Sep-2019. Please note that the revision deadline will expire at 00.00am on this date. If you do not think you will be able to meet this date please let me know immediately.

When submitting your revised manuscript, you will be able to respond to the comments made by the referees and upload a file "Response to Referees" in "Section 6 - File Upload". You can use this

to document any changes you make to the original manuscript. In order to expedite the processing of the revised manuscript, please be as specific as possible in your response to the referees.

on behalf of Dr Jake Socha (Associate Editor) and Kevin Padian (Subject Editor)
openscience@royalsociety.org

Associate Editor Comments to Author (Dr Jake Socha):

Associate Editor: 1

Comments to the Author:

The authors have addressed the major comments of the reviewers, and the manuscript is now acceptable for publication. Before final publication, please address the few remaining minor points of Reviewer 2. Congratulations!

Reviewer comments to Author:

Reviewer: 1

Comments to the Author(s)

Manuscript is acceptable in present form.

Reviewer: 2

Comments to the Author(s)

I thank the authors for their thorough revision of the manuscript.

- I suggest adding some references regarding the trade-off between thermal stress and time spent of the host (subsection 4.1).
- I would also add that using video recordings of landing behavior in the field is absolutely feasible and can be pretty cheap if using simple cameras such as GoPro.
- Finally, several thermographic systems (e.g. FLIR) allow experimenters to calibrate the emissivity during recording and post-recording by ROIs. This has nothing to do with other internal calibration that cameras do during recordings. Depending on the camera you used, your system might not correct for this automatically or might not allow for emissivity adjustments (fixed emissivity): <https://www.jenoptik.us/products/cameras-and-imaging-modules/thermography-camera/handheld-thermographic-camera>. In any case, it seems essential to control for this value and to add it to the material and method section. FYI, by default, the emissivity is often set up at 0.98.

Author's Response to Decision Letter for (RSOS-191119.R1)

See Appendix B.

Decision letter (RSOS-191119.R2)

23-Sep-2019

Dear Dr Horvath,

I am pleased to inform you that your manuscript entitled "Attractiveness of thermally different, uniformly black targets to horseflies: *Tabanus tergustinus* prefers sunlit warm shiny dark targets" is now accepted for publication in Royal Society Open Science.

Kind regards,

Lianne Parkhouse
Royal Society Open Science
openscience@royalsociety.org

on behalf of Dr Jake Socha (Associate Editor) and Kevin Padian (Subject Editor)
openscience@royalsociety.org

Appendix A

Point-by-Point Response to the Comments of Reviewer 1

We thank Referee 1 for her/his constructive and valuable comments. All changes done with respect to the reviews of Referee 1 and Referee 2 are marked with blue and green, respectively. Below one can read our Point-by-Point Response to the points of Referee 1.

Referee 1 wrote: *Page 3, line 19: you should insert the following reference (Perich et al. 1986) after the sentence 'Cattle annoyed by tabanids grow thinner, thus their meat and milk production decreases.'*

Perich M. J., Wright R. E., Lusby K. S. (1986) Impact of horse flies (Diptera: Tabanidae) on beef cattle. Journal of Economic Entomology 79: 128-131

Answer: The requested citation was inserted as follows:

Cattle annoyed by tabanids grow thinner, thus their meat and milk production decreases (Perich et al. 1986).

Perich M. J., Wright R. E., Lusby K. S. (1986) Impact of horse flies (Diptera: Tabanidae) on beef cattle. Journal of Economic Entomology 79: 128-131

Referee 1 wrote: *Page 3, line 59: you should insert the following reference (Krcmar et al. 2014), because of the following reasons: Recently, the efficiency of ten differently colored modified box traps for collecting tabanids was studied in eastern Croatia. In this study the black modified trap was the most successful and collected 20% of all collected tabanids.*

Krcmar S., Radolic V., Lajos P., Lukacevic I. (2014) Efficiency of colored modified box traps for sampling tabanids. Parasite 21: 67 (doi: 10.1051/parasite/2014068)

Answer: The requested citation was inserted as follows:

Field experiments and observations have shown that tabanids prefer sunlit dark hosts/targets (Blahó et al. 2012a, 2013, Egri et al. 2012b, Krcmar et al. 2014).

Krcmar S., Radolic V., Lajos P., Lukacevic I. (2014) Efficiency of colored modified box traps for sampling tabanids. Parasite 21: 67 (doi: 10.1051/parasite/2014068)

Referee 1 wrote: *Page 6, line 5: Identification of horseflies on the experimental plot (barrels) in field experiment with the naked eye on distance of 2 m from the experimental plot (barrels) are very difficult, almost impossible! Tabanus tergstinus is the medium sized horse flies with length from 15 to 18 mm, from distance of 2 m red stripes on green eyes are not visible as stated on the page 8, line 25. Very similar species to T. tergstinus on the first view is Hybomitra ciureai, both species have green eyes with three stripes and reddish-brown sidemarkings on tergites from 1 to 4. The probability of wrong identification is great. I suppose the authors have sampled and identified several species of Tabanus tergstinus by key (Chvala et al. 1972, p. 362-363) before the beginning of the experiment and that on that basis they followed the landing of horseflies on experimental plot (barrels). Following morphological characteristics important to identification of species Tabanus tergstinus should be included in the manuscript. Tabanus tergstinus is the medium sized species with eyes naked and three banded, frons in female narrow, median callus linear and connected with lower callus, thorax is dark*

grey, notopleural lobes dark, abdomen is reddish-brown at sides from one to four tergites and with dark median stripe with indistinct paler median triangles, while posterior tergites uniformly greyish (Chvala et al. 1972).

Chvala M., Lyneborg L., Moucha J. (1972) *The horse flies of Europe (Diptera, Tabanidae)*. Entomological Society of Copenhagen. Copenhagen, Denmark

Answer: As suggested, we changed the first paragraph of the revised subsection 3.4. *Attractiveness of the warm and cold barrels to tabanids* in the following way:

In our field experiment we observed *Tabanus tergustinus* Egger 1859 tabanid flies. During our experiment, on the horse farm's ground there was a rectangular black plastic tray (30 cm × 30 cm) filled with common transparent, yellowish salad oil functioning as an efficient tabanid trap (Egri et al. 2013b). This trap captured only *Tabanus tergustinus* horseflies. They were identified on the basis of the following morphological characteristics (Chvala et al. 1972, p. 362-363): *T. tergustinus* is a medium sized (length 15-18 mm) species with green eyes and three red bands, frons in females are narrow, median callus is linear and connected with lower callus, thorax is dark grey, notopleural lobes are dark, abdomen is reddish-brown at sides from one to four tergites and with dark median stripe with indistinct paler median triangles, while posterior tergites are uniformly greyish. Hence, during our experiment only this tabanid species swarmed in the study area. With the naked eye we could not detect the sex (male or female) of tabanids landed on the barrels, but they might have been females in all probability, because black targets above the ground level are attacked only by female tabanids (Lehane 2005).

Chvala M., Lyneborg L., Moucha J. (1972) *The Horse Flies of Europe (Diptera, Tabanidae)*. Entomological Society of Copenhagen. Copenhagen, Denmark

Referee 1 wrote: Page 10, line 16 to 20: At the end of the sentence which begins 'We suggest... (16 line to 20 line)' you should insert the following:

However, electrophysiological recordings from flight muscle and associated nerves showed that in certain insects the number of nervous impulses impinging on a wing muscle are much fewer than the number of muscle contractions. Thus, the high wingbeat frequency could not be explained on the basis of a one to one relationship between nervous impulse and muscle contraction. While in other cases there was a one to one relationship between nerve impulses and muscle contraction (Romoser and Stoffolano 1998, p. 218).

Romoser W. S., Stoffolano J. G. Jr. (1998) *The science of entomology*. WCB McGraw-Hill Companies, Inc. USA.

Answer: As suggested, in subsection 4.1. *A new hypothesis* we inserted the following:

Blood-seeking tabanids choose preferentially sunlit darker hosts, partly because their wing muscles and neurons function rapidly enough in the warmer microclimate on the body surface of such hosts in order to escape by flying away from the parasite-repelling reactions of the hosts. [Electrophysiological recordings from flight muscle and associated nerves showed that in certain insects the number of nerve impulses impinging on a wing muscle are much fewer than the number of muscle contractions. In this case, high wingbeat frequencies could not be explained on the basis of an one-to-one relationship between nerve impulse and muscle contraction. On the other hand, in other insects there is an one-to-

one relationship between nerve impulses and muscle contractions (Romoser and Stoffolano 1998, p. 218)]. In the colder microclimate on shady darker host animals or sunlit/shady brighter hosts the wing muscles of tabanids may not function rapidly enough for a successful escape.

Romoser W. S., Stoffolano J. G. Jr. (1998) *The Science of Entomology*. WCB McGraw-Hill Companies, Inc. USA.

Point-by-Point Response to the Comments of Reviewer 2

We thank Referee 2 for her/his constructive and valuable comments. All changes done with respect to the reviews by Referee 1 and Referee 2 are marked by blue and green, respectively. Below one can read our Point-by-Point Response to the points of Referee 2.

Referee 2 wrote: *Introduction*

- Remove the page number for the references to Lehane's book (e.g. P2 L8).

Answer: As suggested, all page numbers were removed from the reference (Lehane 2005).

Referee 2 wrote: - *The introduction is rich in references which is important, but some details might be removed so the main message is not diluted (e.g. P3 L2-19).*

Answer: The reference to page numbers by Referee 2 is unclear: Did she/he refer to page numbers in the uppermost row of the pdf file generated by the Royal Society Open Science, or to page numbers in the lowermost row of the original doc manuscript? The relation between these two page numbers is: $N_{pdf} = N_{doc} + 1$. Nevertheless, the following red text (in Lines 2-19 of page $N_{doc} = 2$) was removed together with the corresponding references:

The family Tabanidae consists of more than 4300 species of biting flies belonging to three most important genera *Tabanus* (horseflies), *Chrysops* (deerflies) and *Haematopota* (clegs) (Lehane 2005). These insects are called tabanid flies (or tabanids) further on. A few tabanid species are autogenous, but most require a blood meal for egg maturation (Lehane 2005). Blood-sucking female tabanids are vectors of the pathogens of several dangerous (sometimes lethal) diseases and/or parasites, e.g. tularemia, anaplasmosis, hog cholera, equine infectious anaemia, filariasis, anthrax, Lyme disease and trypanosome (Wiesenhütter 1975, Krinsky 1976, Foil 1989, Luger 1990, Nevill *et al.* 1994, Maat-Bleeker and Bronswijk 1995, Veer *et al.* 2002, Desquesnesab and Dia 2003, Sinshaw *et al.* 2006, Baldacchino *et al.* 2014a, Taioe *et al.* 2017). Their attacks annoy livestock, therefore animals may not graze enough and become thinner. The scars caused by their bites reduce the price of the hide. Some horses attacked by tabanids cannot be ridden, because they throw off the riders (Rutberg 1987, Lin *et al.* 2011). In the tabanid season, a horse may receive up to 4000 bites per day from tabanids with blood loss up to half a litre (Tashiro and Schwardt 1953).

Furthermore, the following red text (in Lines 2-19 of page $N_{doc} = 3$):

Horváth *et al.* (2010) found that owing to the intense tabanid attacks, white and brown horses shuttled between a sunny field and a shady surrounding forest. After a period spent by grazing, the horses escaped from the aggressive tabanids into the shady forest refuge, where they suffered tabanid annoyance only rarely, thus they could rest and wait there quietly. After a certain period, the horses ventured out from the forest shade to graze again in the sunny field, from which they were soon again driven into the forest by tabanids. This shuttling was repeated by the horses periodically until midday (13.00 h), when the tabanid annoyance became so intense that the horses could not graze any further in the field. It was always a brown horse that was first driven into the forest by the attacking tabanids, and it spent 2.2 times longer in the tabanid-free shady forest than in the sunny field. On the other hand, the white horse stayed 1.2 times longer in the sunny field, where it was able to continue to graze, than in the forest. In their field observations, Caro *et al.* (2019) could study the behaviour of tabanid flies around sunlit zebras and horses, because tabanids practically do not attack shady host animals. Otártics

et al. (2019) found that H-traps (composed of a shiny black sphere suspended beneath a tent with an insect-collecting box on its top) placed in sunny places caught significantly more female tabanids than H-traps at shady sites.

was shortened as follows:

Horváth *et al.* (2010) found that in comparison with a white horse, a brown horse spent 2.2 times longer period in a tabanid-free shady forest than in the sunny field with intense tabanid attacks. Caro *et al.* (2019) could study the behaviour of tabanids only around sunlit zebras and horses, because tabanids practically do not attack shady host animals. Otártics *et al.* (2019) found that H-traps in sunny places caught significantly more female tabanids than at shady sites.

Referee 2 wrote: - *I am surprised to see only 1 reference to work conducted on tsetse flies despite the huge amount of literature available on host preference, host seeking (including thermopreference and visual cues) and their similarities in feeding behavior.*

- *The discussion is centered on the hypothesis that the authors make based on their results. It would be interesting to discuss the results in the context of other papers and to expand to other biological systems including tsetse flies. Also, key literature on insects flying at very low temperature including work by B. Heinrich should be included and discussed.*

- *The discussion is also ignoring our current knowledge on host-seeking behavior in blood-sucking insects and the sequence of events regarding the use long-range and short-range cues. I suggest digging into this literature as well.*

Answer: On the one hand, the number of references of our revised manuscript is rather large. On the other hand, the subject of our paper is the thermal preference of tabanids, rather than tsetse flies, the biology, sensory physiology and behaviour of which are rather different from those of tabanids. For example, both male and female tsetse flies suck blood, while in tabanids only the females take blood meals. Both male and female tabanids are positively polarotactic, using linearly polarized light for water detection and host finding, while as far as we know, tsetse flies are not attracted to polarized light. Beyond tsetse flies, many other blood-sucking insects have more or less similarities with tabanids in host preference, host seeking, thermopreference, visual cues and feeding behaviour. These similarities are reviewed in the comprehensive monograph of Lehane (2005), for instance. In our anyway long paper it is not space to summarize these similarities. Thus, in our paper we do not want to deal with any aspect of the biology of tsetse flies and other non-tabanid blood-sucking insects.

Referee 2 wrote: - *P3 L24: a host, if sweating more, will have a bacteria population growing differently compared to a less sweaty host, which can consequently affect chemicals released by the host's skin.*

Answer: As suggested, to the concerned paragraph of the Introduction we added the following:

A more sweating host animal may have a bacteria population growing differently compared to a less sweaty host, which can thus influence chemicals released by the host's skin.

Referee 2 wrote: *M&M*

- P4 L36: *It is unclear whether the warm barrel was filled with water or not. Water vapor is another important cue for host-seeking and if water vapor is emanating from one of the barrels and not this other, that could affect the results.*

Answer: To the revised M&M we added the following:

The warm barrel contained only air which became and remained warm in sunshine, because its closed cap hindered the emanation of warm/hot air. The cold barrel was filled with tap water, into which 10 iceakkus (Aspico G40, 0.25 litre, 0.76 kg plastic container filled with a gel of low temperature of congelation) frozen in a common kitchen-refrigerator were submerged. Its closed cap hindered the outflow of the cold water or the emanation of water vapour.

Referee 2 wrote: - *I am concerned about the humidity on the cold barrel due to condensation that could affect the vision of the fly.*

- P5 L50-54: *I have a hard time imagining that a barrel containing water at 0°C would not create condensation at ambient temperature. What was the temperature of the environment during the experiments? What about relative humidity?*

Answer: Although the water temperature in the cold barrel was about 0 °C at the beginning and remained relatively cold during our experiments, the temperature of the antisolar (shady) outer surface of the cold barrel was much higher (20-30 °C, see Figure 1), because the plastic barrel's wall (with 5 mm thickness) was a good thermal insulator. Therefore, the outer surface temperature was never so cold that the air humidity could have been condensed on the wall. In subsection 2.2. *Imaging polarimetry* of our revised manuscript we have written the following:

Only tiny water droplets condensed from air humidity on the outer surface of the cold barrel could have resulted in some differences in reflection-polarization, but such a condensation did not occur in our field experiments. The temperature of the antisolar outer surface of the cold barrel was between 20 and 30 °C (Fig. 1), because the plastic barrel's wall with 5 mm thickness was a good thermal insulator. Therefore, the outer surface temperature was never so cold that the air humidity could have been condensed on the wall.

Referee 2 wrote: - *Did you monitor the ambient temperature during your experiments?*

Answer: The air temperature was not continuously monitored during our experiments. We have measured half hourly/hourly only the surface temperatures of the warm and cold barrels with a thermocamera, because they are the most important variables determining the landing and staying reactions of tabanids on the barrels. To subsection 2.3. *Thermography* of the revised M&M we added the following:

The air temperature was not continuously monitored during our experiments, but it changed between about 28 and 36 °C from the beginning to the end of the experiments.

Referee 2 wrote: - P4 L48-L54: *I would remove the results from exp. 1 from the analysis then.*

P5: Based on the issues that arose during experiment 1 and the fact that the number of observers was different between the experiments, I strongly recommend removing data for exp 1 and 2 for the final analysis. This will make the results cleaner and more replicable.

Answer: On the one hand, the data collected in experiments 1 and 2 are important, because our main aim was to determine the number N of landings and the average time period $\langle t \rangle$ spent by tabanids on the target surface as a function of the surface temperature T (Figure 7). For this aim many different T -values of the barrel's surface were necessary. It was all the same that these data were collected by one experimenter observing only the solar sides of the warm and cold barrels, or by two experimenters observing simultaneously both solar and antisolar sides. In subsection 2.1. *Choice experiments* of the revised Materials and Methods we wrote:

'Since we did not want to lose the results of experiment 1, we used these data, although the temporal temperature variation of the water-filled cold barrel was different in the first experiment (without refreshing the iceakkus) compared to the other four experiments (with refreshing the iceakkus). The only irrelevant consequence of this was that in the second half of experiment 1 higher surface temperatures of the water-filled barrel occurred than in experiments 2-5. This, however, did not influence our results and conclusions: **Our main aim was to determine the number of landings and the average time period spent by tabanids on target surfaces as a function of the surface temperature T . For this aim many different T -values of the barrel's surface were necessary. It was all the same that these data are collected by one experimenter observing only the solar sides of the warm and cold barrels, or by two experimenters observing simultaneously both solar and antisolar sides.'**

On the other hand, the statistical analyses were performed separately for experiments 1-2 and experiments 3-5 (see Figures 5 and 6).

Finally, both experiment types with 1 or 2 experimenters can be easily replicated.

Thus, based on these arguments, we do not want to remove the important results of experiments 1 and 2.

Referee 2 wrote: - P5: *Please describe more the behavior of the experimenters during the observations. Were they sitting or walking around the barrels?*

Answer: To subsection 2.1. *Choice experiments* of the revised M&M we added:

These observers wore white clothes and hats against direct sunshine **and to minimize their visual attractiveness to tabanids. They were sitting on a chair during the observations.**

Referee 2 wrote: - P5 L17-22: *how did you count the number of landing? Did you use a counter? Why not using video recordings? That would have allowed you to avoid any bias associated with the experimenters being around the tank, including the fact that they released CO2 and other body odors, humidity and heat (all cues used by these flies to locate a host).*

Answer: An automatic continuous video recording during our field experiments was technically not possible. This would have required the following enormously difficult logistics: (i) Electric power supply of the two video cameras monitoring simultaneously the solar and antisolar sides of the sunlit

barrels for daily 8 hours. (ii) Automatic hourly turning of the connecting line of the center of the two barrels together with their tetrapodal plastic stools and the two monitoring cameras in such a way that the line's normal vector points toward the sun to ensure that the sunlight hitting the barrels is always nearly parallel to the direction of observation. (iii) Automatic hourly inversion of the positions of the warm and cold barrels together with their tetrapodal plastic stools during each experiment in order to avoid site bias. (iv) Automatic refreshment of the iceakkus in the cold barrel at half-time of each experiment. Such logistics could be performed only by the National Aeronautics and Space Administration (NASA) during a Mars mission, for example. ☺

In subsection 2.1. *Choice experiments* of the revised M&M we wrote:

During observations, we counted with a 5-minute temporal resolution how many landings were on the solar/antisolar sides of the warm/cold barrels, and how many seconds the landed tabanids spent on them. **Counting happened with the naked eye, and time periods were measured with a stopwatch.**

Referee 2 wrote: - P5/P6: *thermography with horses: how many measurements did you do? How many horses did you use for your measurements?*

Answer: In subsection 2.3. *Thermography* of the revised M&M we wrote:

The distributions of the surface temperature (thermograms) of **two (1 sunlit and 1 shady) white, two brown and two black horses** as well as the two black barrels used in our field experiments were measured with an infrared camera (VarioCAM[®], Jenoptik Laser Optik Systeme GmbH, Jena, Germany). Horses were measured under sunny (illuminated with direct sunlight) and shady (when the sun was behind a cloud) conditions. **At a given (sunlit or shady) horse we recorded several thermograms and selected a typical thermogram where the optical axis of the thermocamera was nearly perpendicular to the long axis of the horse.**

Referee 2 wrote: - P5/P6: *thermography with horses: What value was the emissivity set up at?*

Answer: We do not understand this question. What does mean Referee 2 on 'setting up the emissivity value'?

Our thermocamera (VarioCAM[®], Jenoptik Laser Optik Systeme GmbH, Jena, Germany) measures the surface temperature with a nominal precision of ± 1.5 °C and is equipped with a so-called internal „shutter calibration” mechanism ensuring continuous precise temperature measurement: During a shutter calibration, a blackbody is slid in the field of view within the camera's enclosure. Knowing the temperature of this blackbody, which is assumed to be the same as the internal temperature (electronically maintained at around 30 °C), the system calculates and adjusts the gain and offset for each pixel. Under normal conditions of use, a shutter calibration is automatically performed every minute resulting in an effective measurement accuracy of ± 1.5 °C with a temperature resolution of better than 0.07 °C. This means that a „true” target temperature at a given point (measured e.g. by an accurate contact thermometer) might have an offset of ± 1.5 °C on the IR picture. Nevertheless, the high thermal resolution determining the differences between neighbouring pixels results in an accurate representation for *temperature differences* in a given snapshot. Note furthermore that the colour scale has nothing to do with temperature reading, data are stored as calibrated temperature values for each pixel.

Thus, our thermocamera does not need any manual ‘setting up of the emissivity value’.

Referee 2 wrote: - P6: Why testing turbulent / windy conditions if tabanids avoid flying under such conditions?

Answer: In subsection 3.2. *Schlieren imagery of the warm/cold air next to the warm/cold barrels* of the revised Results we wrote:

Note that tabanids also fly in weaker winds, when the air flow is turbulent around any target (host animals or our barrels).

Referee 2 wrote: Results

- Figure 4: the scale is too large to notice any difference in temperature across the body of the horses. Also, related to P8 L60: the surrounding area (unless you changed the emissivity for these specific ROIs) might not be warmer or cooler, depending on the specific emissivity of the material.

Answer: In Figure 4 we used the same temperature scale (from T_{\min} to T_{\max}) for all six (sunlit and shady black, brown, white) horses, because their colour-coded temperatures should be comparable. Using different temperature scales for different horses would have resulted in ambiguousness: a given colour would code different temperatures, and *vice versa*, a given temperature would be coded by different colours. We did not want such an ambiguous temperature colour coding.

On the other hand, the surrounding area of the studied horses can be warmer or cooler than the surface temperature of horses, depending on the albedo/colour of the surface and the angle between the local direction of the normal vector of the surface and the sunrays. In the original Discussion we wrote: ‘*In the top-left (sunlit black horse) and bottom-right (shady white horse) thermograms of Fig. 4 the temperature of ground with heterogeneous albedo and thermophysical characteristics was approximately 60-65 °C. These hot patches correspond to dry soil regions or dung piles, which warmed up very much in direct sunshine.*’

This statement is correct for the following reasons:

In the original site (on a horse farm) of our field experiments, on 10 June 2019 under sunny conditions with 29 °C air temperature, we have measured the soil surface temperature by a contact thermometer (GAO Digital Multitester EM392B 06554H, EverFlourish Europe GmbH, Friedrichsthal, Germany) with nominal precision of ± 1 °C, and registered the thermograms of different sunlit surface patches by our thermocamera (VarioCAM[®], Jenoptik Laser Optik Systeme GmbH, Jena, Germany) with nominal precision of ± 1.5 °C. Response-Figure 1 illustrates three examples for ground surface temperatures of 64, 72 and 78 °C. Response-Figure 2 presents examples for ground surface temperatures of 63 and 77 °C measured simultaneously by contact thermometer and thermocamera. All these data provide experimental evidence that the sunlit ground surface temperature can be as warm as 60-65 °C and even much higher in the site of our original field experiments.

Response-Figure 1: Examples for ground surface temperatures of 64, 72 and 78 °C measured by contact thermometer (with the metal tip at the end of its electrically insulated green wire) at the site of our field experiments (photos are taken by Gábor Horváth).

Response-Figure 2: Examples for ground surface temperatures of 63 and 77 °C measured simultaneously by contact thermometer and thermocamera at the site of our field experiments (photos and thermograms are taken by Gábor Horváth and Imre János).

Referee 2 wrote: - *Figure 7: Is this data related to the warm barrel cooling down or the cool barrel warming up (or both)? Any stats / regression for this data?*

Answer: We wrote in subsection 2.3. *Thermography* of the original Materials and Methods (see also Fig. 2C,D): ‘*The thermograms of the sunlit and shady sides of barrels were measured periodically (half hourly or hourly) in full sunshine under cloudless sky from the beginning to the end of each experiment. ... On the barrel’s sides an uniform rectangular area was selected, where $\langle T \rangle$ and ΔT was determined.*’

To the legend of Figure 7 we added the following:

All temperature datas originate from the thermographically measured, temporally changing surface temperatures of the solar and antisolar sides of the warm and cold barrels shown in Fig. 1.

We wrote in the original caption for Fig. 7: ‘The straights show the regression line fitted to the data points represented by symbol +. The grey area around the regression line shows 95 % confidence interval.’

The regression lines and their 95 % confidence intervals were enough to clearly show the tendencies (i.e. N and $\langle t \rangle$ increased with increasing T) and to draw correct conclusions.

Referee 2 wrote: - Table S3: did you use a statistical correction to adjust for multiple comparisons?

Answer: To draw our conclusions, it was not necessary to do any statistical correction on our data. If, however, Referee 2 explicitly writes us which statistical correction would be necessary and why, we shall perform it.

Referee 2 wrote: - Why so many flies landed on the cold barrel? You mentioned that no flies landed on the barrel when $T > 18$ °C.

Answer: On the one hand, Referee 2 cited erroneously our finding: We found that if T was smaller than 18 °C (hence $T < 18$ °C, rather than $T > 18$ °C !), tabanids did not land on the barrels.

On the other hand, tabanids cannot sense remotely the temperature of a surface. They can sense it only after landing. Thus, the logical expectation was that nearly as many landings will happen on the warm barrel as on the cold barrel, both being optically identical, and thus, visually uniformly attractive to tabanids. However, in our experiments we found that on the warm barrel significantly more tabanids landed than on the cold barrel. Consequently, the logical question should be: Why less tabanid landings happened on the cold barrel than on the warm one, in spite of the fact that they were optically identical? The answer was given in subsections 4.1. and 4.2. as follows:

4.1. A new hypothesis:

In our opinion, both the warm/cold boundary layer of a few centimeters and the warm/cold turbulent air train of a few decimeters next to the target’s wall (Fig. 3) are important to explain the behaviour of tabanids flying around the barrels in our field experiments: within a few to several centimeters, flying tabanids may sense the temperature of the boundary layer (microclimate) and thus could estimate the temperature of the barrel’s surface and decide whether they do or do not alight on it.

4.2. Fullfilment of the prerequisite of the new hypothesis:

Since a remotely thermosensitive organ detecting infrared radiation by tabanids has not been discovered, the only explanation of our finding is that prior to alighting, the tabanids flying close around the barrels sense the colder air (boundary) layer next to the vertical wall of the colder barrel. This boundary layer was not warm enough, thus tabanids landed only rarely on the colder barrel, and if they looked for blood, they remained only for a very short periods on the cold substratum.

Referee 2 wrote: *It is unclear to me how the temperature of the barrel changed over time and why it was not maintained at one given T for the whole tests.*

Answer: The temporal change of the surface temperature of the solar and antisolar sides of the warm and cold barrels is clearly shown in Figure 1.

To keep the surface temperature of sunlit (air- and water-filled) barrels constant would have been impossible, because the incident angle of sunlight changed continuously during our experiments due to the changing elevation of the sun. When the incident angle (from the barrel's surface) of sunlight decreased/increased, the barrel's surface temperature decreased/increased. This change should have been somehow compensated with an appropriate change of the inner air/water temperature of the barrels (e.g. with thermistor-controlled warming/cooling), if the surface temperature should have been constant. Furthermore, sometimes the sun became occluded by clouds, when the surface temperature of the barrels decreased. This should have also been compensated somehow.

We deliberately did not maintain the surface temperature T of barrels at a constant value, because changing T (see Figure 1) offered many different temperature values for landing tabanids, and thus we could study the number of landings and the average time period spent on the surface as a function of T .

Referee 2 wrote: *If T changed over time, then it is affecting your statistical analyses unless you first create categories / ranges and then compare them.*

Answer: The changing T did not affect our statistical analyses (which were performed only in Figures 5-6 and Supplementary Table S3), because one of our aims was to determine the T -dependence of the number N of landings and the average time period $\langle t \rangle$ spent by tabanids on the barrel's surface (see Figure 7). In the case of the $N(T)$ and $\langle t \rangle(T)$ datas, the regression lines and their 95 % confidence intervals are important, rather than any statistics.

Referee 2 wrote: *Discussion*

- P8: The first part of the discussion is useless (L4-45) and should be removed along with data from experiments 1 and 2 as stated before. That would make this paper and its take home message stronger. Moreover, some of these are overstatements or just impossible to verify (e.g. the CO₂ released by the two observers was similar and did not affect the experiments).

Answer: As suggested, the following text was removed from the revised Discussion:

The persons observing the two black barrels in our field experiments wore white clothes and hats against direct sunshine. It is well-known that tabanids are not attracted to white targets (Bracken *et al.* 1962, Granger 1970, Roberts 1970, Browne and Bennett 1980, Allan and Stoffolano 1986, Sasaki 2001, Lehane 2005, Horváth *et al.* 2008, 2010, Kriska *et al.* 2009, Blahó *et al.* 2012a, Egri *et al.* 2013a, Horváth 2014). Thus, the presence of these observers 2 m from the barrels could only minimally (if at all) attract horseflies.

Since tabanids are attracted to CO₂ (Roberts 1977, Hall *et al.* 1998, Krcmar 2005, 2013, Lehane 2005, Mihok and Mulye 2010, Mihok and Lange 2012, Blahó *et al.* 2013, Baldacchino *et al.* 2014), the carbon dioxide exhaled by the observers could be an olfactorial bait in the context of the barrels as visual targets. This could account for some of the scatter in our temperature versus landing graphs shown in Figs. 5, 6 and 7. However, the CO₂ exhaled by the observers could influence the attractiveness of our barrels to tabanids only minimally for the following three reasons:

(i) According to our earlier field experiments performed at the same site where an automatic meteorological station functioned continuously from the beginning of June to the end of September 2018 (Horváth *et al.* 2018), the direction of local wind speed at the experimental site varies randomly during the day in summer. Thus, the exhaled CO₂ reached the barrels from random directions.

(ii) The positions of the warm and cold barrels were hourly inverted during our experiments. The possible influence of the exhaled CO₂ on the tabanid-attractiveness of barrels was minimized by these alternating positions of the barrels.

(iii) The direction of observation was perpendicular to the connecting line passing through the center of the barrels. The connecting line between the barrels was hourly changed in such a way that its normal vector pointed toward the sun. Also by this we minimized the possible bias in the tabanid-attractiveness of barrels due to the CO₂ exhaled by the observers.

(iv) The observers were equally 2 m from the two barrels; this symmetry also minimized the possible bias induced by the exhaled carbon dioxide.

Tabanids might have landed more often and/or stayed longer when the observers were downwind/upwind of a given barrel as opposed to crosswind. However, this possibility was practically eliminated by the randomly varying wind direction and the hourly changing positions and relative orientations of our barrels mentioned in points (i)-(iv).

In our field experiments, the number of tabanids landed on our barrels could simply be enhanced by flies being attracted to where greater numbers of other flies are and this could be a possible confound. However, in a given time period there was always only one tabanid individual on a barrel. This observational fact evidently excluded the possibility that a second tabanid might have been landed on the same barrel due to the presence of the first tabanid.

Referee 2 wrote: - *P9: The flight muscles of these insects probably don't really cool down once they are on the host because of 1) their ability to feed extremely fast which help them stay warm and 2) their ability to maintain these muscles by shivering. You would need thermographic analyses to support your hypothesis regarding the cooling of these muscles for example or thermocouple recordings.*

Answer: Yes, it would be worth performing thermographic analyses and thermocouple recordings from the wing muscles of tabanids landed on sunlit and shady living horses of different colours. These could be the task of future research. Note, however, that with thermography only the temperature of the chitinous thorax surface of a tabanid fly could be measured, rather than that of the underlying wing muscles. For such a thermography the thorax wall should be open in order that the thermocamera can see the wing muscles. This can be performed only in a laboratory with fixed and prepared tabanids. It is questionable how the real environmental conditions of tabanid landing on the body surface of a real horse could be ensured in a laboratory? Similar problems occur during the mentioned thermocouple recordings from tabanid wing muscles.

Referee 2 wrote: - *Feeding on a warmer host presents the advantage of being able to feed faster and thus minimizing the time spent of the host and consequently minimizing the hosts' anti-parasitic behaviors. This has a potential cost too: thermal stress. Another advantage is that feeding faster minimizes the decrease in body temperature of the insect. This should be discussed and included here.*

Answer: Thanks, both sound ideas were inserted in subsection 4.1. A new hypothesis of the revised Discussion as follows:

Feeding on a warmer host may have the advantage that tabanids could feed faster (e.g. because of the faster functioning of the musculature of their mouth part) with which they could minimize the time

spent on the host, and thus minimize the hosts' anti-parasitic behaviours. However, this has the potential cost of thermal stress. Another advantage could be that feeding faster minimizes the decrease in the temperature of the insect's body, inclusive wing muscles. Note that both advantages are also related to the faster functioning of tabanid muscles.

Appendix B

Point-by-Point Response to the new Comments of Reviewer 2

We thank Referee 2 for her/his new comments. All changes done with respect to the new review by Referee 2 are marked by green. Below one can read our Point-by-Point Response to the new points of Referee 2.

Referee 2 wrote: *I suggest adding some references regarding the trade-off between thermal stress and time spent on the host (subsection 4.1).*

Answer: As suggested by Referee 2, in the revised subsection '4.1. A new hypothesis' we cited references [4, 32, 64, 65] in the following way:

Feeding on a warmer host may have the advantage that tabanids could feed faster (e.g. because of the faster functioning of the musculature of their mouth part [32]) with which they could minimize the time spent on the host, and thus minimize the hosts' anti-parasitic behaviours [4]. However, this has the potential cost of thermal stress [65]. Another advantage could be that feeding faster minimizes the decrease in the temperature of the insect's body, inclusive wing muscles [64]. Note that both advantages are also related to the faster functioning of tabanid muscles.

4. Lehane MJ. 2005 *The biology of blood-sucking in insects*. 2nd ed, Cambridge, UK: Cambridge University Press
32. Elzinga RJ. 2003 *Fundamentals of entomology*, 6th ed. Upper Saddle River, NJ: Prentice Hall
64. Romoser WS, Stofolano JG Jr. 1998 *The science of entomology*. New York: WCB McGraw-Hill Companies, Inc.
65. Moyes CD, Schulte PM. 2014 *Principles of animal physiology*. 2nd ed., Harlow, Essex, UK: Pearson Education Ltd.

Referee 2 wrote: *I would also add that using video recordings of landing behavior in the field is absolutely feasible and can be pretty cheap if using simple cameras such as GoPro.*

Answer: As suggested by Referee 2, in the revised 'Introduction' section we wrote:

Using video recordings of tabanids in the field, Caro *et al.* [43] could study the behaviour of horseflies only around sunlit zebras and horses, because tabanids practically do not attack shady host animals.

Referee 2 wrote: *Finally, several thermographic systems (e.g. FLIR) allow experimenters to calibrate the emissivity during recording and post-recording by ROIs. This has nothing to do with other internal calibration that cameras do during recordings. Depending on the camera you used, your system might not correct for this automatically or might not allow for emissivity adjustments (fixed emissivity): <https://www.jenoptik.us/products/cameras-and-imaging-modules/thermography-camera/handheld-thermographic-camera>. In any case, it seems essential to control for this value and to add it to the material and method section. FYI, by default, the emissivity is often set up at 0.98.*

Answer: As suggested by Referee 2, in the revised 'Material and methods' section we wrote:

The correctness of the temperature measurement of our thermocamera was validated by a calibration procedure with a contact thermometer (GAO Digital Multitester EM392B 06554H, EverFlourish Europe GmbH., Friedrichsthal, Germany) with a nominal precision of ± 1 °C. Further details about this calibration can be read in the Electronic Supplementary Materials (Supplementary Figs. S1-S3).

Furthermore, to the revised Electronic Supplementary Materials we added the following:

Supplementary Materials and Methods

Calibration of the Thermocamera

In order to demonstrate that our thermocamera (VarioCAM[®], Jenoptik Laser Optik Systeme GmbH, Jena, Germany, with a nominal precision of ± 1.5 °C) really measures the surface temperature correctly and precisely, we have measured the surface temperature of a cold and warm black coffee between 3 and 55 °C (Supplementary Figure S1) and a hot reddish tile between 27 and 67 °C (Supplementary Figure S2) by our thermocamera and contact thermometer (GAO Digital Multitester EM392B 06554H, EverFlourish Europe GmbH., Friedrichsthal, Germany, with a nominal precision of ± 1 °C). Supplementary Figure S3 displays the graph of surface temperatures measured by our thermocamera and contact thermometer.

These data provide experimental evidence that our thermocamera can measure the surface temperature with an accuracy of about ± 1.5 °C.

The small differences between the temperatures measured simultaneously by the contact thermometer and the thermocamera are due to many factors: (i) The metal tip of the contact thermometer conducted some heat flux from/to the air to/from the tiny target volume to be measured. (ii) This tip was immersed in the coffee slightly (circa 1 mm) below the coffee-air interface, thus it could not measure the temperature of the coffee surface, which was measured by the thermocamera. (iii) At higher temperatures, the enhanced evaporation strongly cools the fluid surface (skin layer) resulting in systematically increasing differences between the two temperature values. (iv) The majority of the tip of the contact thermometer was in the air layer directly above the tile's surface. Obviously, the contact thermometer could not be pressed into the uppermost thin (20-50 μm) surface layer of the tile from which the electromagnetic radiation was emitted.

thermocamera

contact thermometer

Supplementary Figure S1: Thermograms of a coffee surface with different temperatures ranging from 3 to 25 °C measured simultaneously by our thermocamera and contact thermometer (photos and thermograms are taken by Gábor Horváth and Imre János).

thermocamera

contact thermometer

Supplementary Figure S1 continued: Thermograms of a coffee surface with different temperatures ranging from 26 to 55 °C (photos and thermograms are taken by Gábor Horváth and Imre János).

Supplementary Figure S2: Thermograms of the surface of a reddish tile with different surface temperatures ranging from 27 to 67 °C measured simultaneously by our thermocamera and contact thermometer (photos and thermograms are taken by Gábor Horváth and Imre János).

Supplementary Figure S3: Graph of surface temperatures measured simultaneously by our thermocamera (T_t : vertical axis) and contact thermometer (T_c : horizontal axis) (see Supplementary Figures S1 and S2). The dashed line corresponds to $T_t = T_c$.